# The severity of microstrokes depends on local vascular topology and baseline perfusion

Franca Schmid[1,2]*, Giulia Conti[2], Patrick Jenny[2], Bruno Weber[1,3]*

[1]Institute of Pharmacology and Toxicology, University of Zurich, Zurich, Switzerland; [2]Institute of Fluid Dynamics, ETH Zurich, Zurich, Switzerland; [3]Neuroscience Center Zurich, University and ETH Zurich, Zurich, Switzerland

**Abstract** Cortical microinfarcts are linked to pathologies like cerebral amyloid angiopathy and dementia. Despite their relevance for disease progression, microinfarcts often remain undetected and the smallest scale of blood flow disturbance has not yet been identified. We employed blood flow simulations in realistic microvascular networks from the mouse cortex to quantify the impact of single-capillary occlusions. Our simulations reveal that the severity of a microstroke is strongly affected by the local vascular topology and the baseline flow rate in the occluded capillary. The largest changes in perfusion are observed in capillaries with two inflows and two outflows. This specific topological configuration only occurs with a frequency of 8%. The majority of capillaries have one inflow and one outflow and is likely designed to efficiently supply oxygen and nutrients. Taken together, microstrokes bear potential to induce a cascade of local disturbances in the surrounding tissue, which might accumulate and impair energy supply locally.

*For correspondence:
frschmid@ethz.ch (FS);
bweber@pharma.uzh.ch (BW)

**Competing interests:** The authors declare that no competing interests exist.

## Introduction

As the brain's energy storage is limited, a sustained supply of oxygen and nutrients is crucial to avoid local tissue damage. Accordingly, flow disturbances even at the level of individual vessels can result in cortical tissue lesions, so called microinfarcts (*Nishimura et al., 2007*; *Shih et al., 2013*; *Zhang et al., 2015*; *Taylor et al., 2016*; *Summers et al., 2017*). In recent decades, it has become evident that such microinfarcts are linked to various pathologies, for example cerebral amyloid angiopathy (CAA), Alzheimer's disease (AD), and dementia (*van Veluw et al., 2021*; *Shih et al., 2018*; *Pétrault et al., 2019*; *van Veluw et al., 2017*; *Smith et al., 2012*). Depending on the severity of flow disturbance, the dimension of documented microinfarcts ranges between 50 μm to a few millimeters (*Shih et al., 2013*; *Taylor et al., 2016*; *Summers et al., 2017*; *Shih et al., 2018*; *van Veluw et al., 2017*; *Smith et al., 2012*; *Brundel et al., 2012*). This small size renders the quantification of the brain's microinfarct burden challenging and makes it difficult to gain insights on their role for local blood supply.

Animal models allow a more refined study of the etiology of microinfarcts (*Shih et al., 2018*). By occluding vessels via photothrombosis (*Nishimura et al., 2007*; *Shih et al., 2013*; *Zhang et al., 2015*; *Taylor et al., 2016*; *Summers et al., 2017*; *Underly et al., 2017*; *Nishimura et al., 2006*; *Zhang et al., 2020*; *Nishimura et al., 2010*; *Enright et al., 2007*) or by injecting microemboli (*Wang et al., 2017*; *Wang et al., 2012*; *Nozari et al., 2010*; *Silasi et al., 2015*; *Zhu et al., 2012*; *Lam et al., 2010*), microstrokes can be induced and their impact on blood flow and on the surrounding tissue can be studied. Here, most studies focus on the occlusion of penetrating vessels, which because of their one-dimensional topology (*Schmid et al., 2019a*; *Duvernoy et al., 1981*; *Blinder et al., 2013*) have been identified as the 'bottleneck of perfusion' (*Nishimura et al., 2007*). Less attention has been given to occlusions of descending arteriole (DA) offshoots and capillaries.

**eLife digest** A blockage in one of the tiny blood vessels or capillaries of the brain causes a 'microstroke'. Microstrokes do not cause the same level of damage as a major stroke, which is caused by a blockage in a larger blood vessel that completely cuts off oxygen to a part of the brain for a period. But microstrokes do increase the risk of developing conditions like dementia – including Alzheimer's disease – later in life.

People with these neurodegenerative conditions have fewer capillaries in their brains. The capillaries make up a mesh-like network of millions of vessels that supply most of the energy and oxygen to the brain. Repeated microstrokes may contribute to progressive loss of capillaries over time. Reduced numbers of capillaries may increase memory loss and other brain difficulties.

To better understand how microstrokes affect blood flow in the brain, Schmid et al. created a computer model to simulate blood flow in capillaries in the mouse brain. Then, they modeled what happens to the blood flow when one capillary is blocked. The experiments showed that the configuration of the blocked capillary determines how much blood flow in neighboring capillaries changes. Blockages in capillaries with two vessels feeding in and two vessels feeding out caused the greatest blood flow disturbances. But these 2-in-2-out vessels only make up about 8% of all brain capillaries. Blockages in capillaries with different configurations with respect to feeding vessels had less effect.

The experiments suggest that most microstrokes have limited effects on blood flow on the scale of the entire brain because of redundancies in the capillary network in the brain. However, the ability of the capillary network to adapt and reroute blood flow in response to small blockages may decrease with aging. Over time, ministrokes in a single capillary may set off a chain reaction of disturbed blood flow and more blockages. This may decrease energy and oxygen supplies explaining age- and disease-related brain decline. Better understanding the effects of microstrokes on blood flow may help scientists develop new ways to prevent such declines.

While the occlusion of DA offshoots causes a maximal infarct volume of 0.8 nl (275 times smaller than for DA occlusions), no tissue damage could be detected for the occlusion of capillaries > 2 branches apart from the DA (*Shih et al., 2013*). However, the effect of anesthesia on these results remains unknown. This is because anesthesia can act as a vasodilator and tends to increase red blood cell (RBC) flux and tissue oxygenation (*Lyons et al., 2016*; *Roche et al., 2019*). Importantly, it has also been shown that single-capillary occlusion causes flow reversals and RBC speed reductions of up to 90% in the vessels downstream of the occluded capillary (*Nishimura et al., 2006*). Additionally, singlecapillary occlusion can induce the formation and alter the morphology of amyloid-beta (Aβ) plaques (*Zhang et al., 2020*), which are related to AD and CAA. Taken together, even if single-capillary occlusion might not directly cause local tissue damage, it disturbs blood flow locally and might impair tissue clearance. As such, single-capillary occlusions could play an important role in the development and progression of larger disturbances and pathologies.

Further studies have investigated the effect of simultaneously occluding multiple capillaries or multiple microvessels of larger caliber (*Underly et al., 2017*; *Lam et al., 2010*). *Underly et al., 2017* showed that the occlusion of ~10 proximal capillaries via photothrombosis leads to deterioration of the blood brain barrier (BBB). The accumulation of occluded capillaries can also influence the total perfusion of the cortical vasculature. In a simulation study related to in vivo investigations in an AD mouse model, *Cruz Hernández et al., 2019* showed that with 2% of capillaries stalled, cortical cerebral blood flow is reduced by ~5%. Capillary stalls have also been identified as an important factor for incomplete reperfusion after stroke (*El Amki et al., 2020*; *Erdener et al., 2021*).

These aspects underline the need for an in-depth quantification of blood flow changes in response to single-capillary occlusion, which will allow us to better understand the role of these small disturbances on local tissue perfusion. We will identify factors influencing the severity of micro-occlusions and quantify the area of impact. Additionally, by looking at the smallest possible scale of occlusion, valuable insights on the robustness of perfusion within the capillary bed can be gained and our knowledge of topological characteristics of cortical capillary beds can be extended. In this context, we will also analyze the arrangement of arteriole- and venule-sided capillaries. Moreover, identifying

the smallest scale of disturbance is a prerequisite for the correct interpretation of disturbances and changes observed on a larger scale.

To address these questions, we employed blood flow simulations in realistic microvascular networks (MVNs) from the mouse cortex (*Blinder et al., 2013*; *Schmid et al., 2017*; *Schmid et al., 2019b*), in which individual capillaries have been occluded. Using an in silico approach comes with several advantages. First of all, it is challenging to monitor blood flow changes in vivo with single vessel resolution in multiple vessels or even entire vascular networks simultaneously. This problem is even more pronounced, if the focus is on blood flow changes in the capillary bed, which is highly interconnected (*Schmid et al., 2019a*; *Blinder et al., 2013*; *Schmid et al., 2017*; *Lauwers et al., 2008*; *Hirsch et al., 2012*; *Smith et al., 2019*) and in which the flow field is highly heterogeneous and fluctuating (*Schmid et al., 2017*; *Schmid et al., 2019b*; *Kleinfeld et al., 1998*; *Villringer et al., 1994*; *Guibert et al., 2010*; *Parpaleix et al., 2013*). Secondly, in silico studies allow us to investigate the impact of single-capillary occlusions in an isolated manner. This is in contrast to in vivo analyses, where a capillary occlusion will always be accompanied by a response from directly neighboring cells (e.g. endothelial cells, mural cells, microglia).

By studying flow changes in response to the occlusion of 167 different capillaries, we reveal that the severity of a microstroke strongly depends on the local vascular topology and the baseline flow rate in the occluded capillary. More precisely, in the worst-case scenario, the flow rate dropped by as much as 70% in the direct vicinity of the occluded capillary. As well as this, a microstroke locally reduces the number of available flow paths between DAs and ascending venules (AVs). This aspect might play an important role for the upregulation of blood flow during neural activation, since the ability of the microvasculature to adapt to local changes in energy demand might be impaired in the disturbed flow field. The fact that the worst-case scenario only occurs with a frequency of 8% across all capillaries and the re-routing of blood to neighboring vessels suggests that the capillary bed offers an inherent robustness toward single-capillary occlusions. These aspects as well as compensatory oxygen supply from neighboring capillaries likely help to avoid severe hypoxic conditions in response to single-capillary occlusion. Our results further indicate that the different vascular topologies are not only relevant for the severity of the microstroke, but that they might in fact fulfill distinct functional tasks. We postulate that there is a topological difference between capillaries responsible for the distribution of blood and capillaries responsible for supplying oxygen and nutrients to the cortical tissue.

In summary, our work provides an in-depth quantification of flow changes in response to single-capillary occlusions and reveals novel topological characteristics of the cortical microvasculature. Our results give valuable insights into the role of microinfarcts, which are relevant for future in vivo studies on the robustness of oxygen and nutrient supply and on pathological flow disturbances.

## Results

The following results are based on time-averaged blood flow simulations in realistic MVNs embedded in a tissue volume of 1.6 mm³ (MVN1) and 2.2 mm³ (MVN2). The realistic MVNs and the simulation framework have been introduced in previous publications (*Blinder et al., 2013*; *Schmid et al., 2017*; *Schmid et al., 2015*). A brief description of both is provided in Materials and methods. In total, we performed 167 single microstroke simulations to investigate the impact of a microstroke on local perfusion and to reveal key factors for the severity of the microstroke. To induce a microstroke, we constricted the diameter of the microstroke capillary (MSC) to 0.01 µm, which reduced the flow rate in the MSC to $< 10^{-10} \mu m^3 m s^{-1}$. Details regarding the selection of MSCs and the computation of the relative flow changes $\Delta q_{ij}$ are provided in the Materials and methods and *Supplementary file 1a*. Statistical validations are available in *Supplementary file 1c–e*.

### The severity of a microstroke is governed by the local vascular topology

Microvascular bifurcations are either divergent or convergent. Thus, depending on the bifurcation types at the source and the target vertex of the MSC, four topological configurations are possible at the MSC (*Figure 1a–d*). To identify the MSC type, the flow directions in all five capillaries need to be known. An identification based purely on the topological arrangement and appearance of the vessel is not possible. To investigate the impact of the topological configuration on the severity of

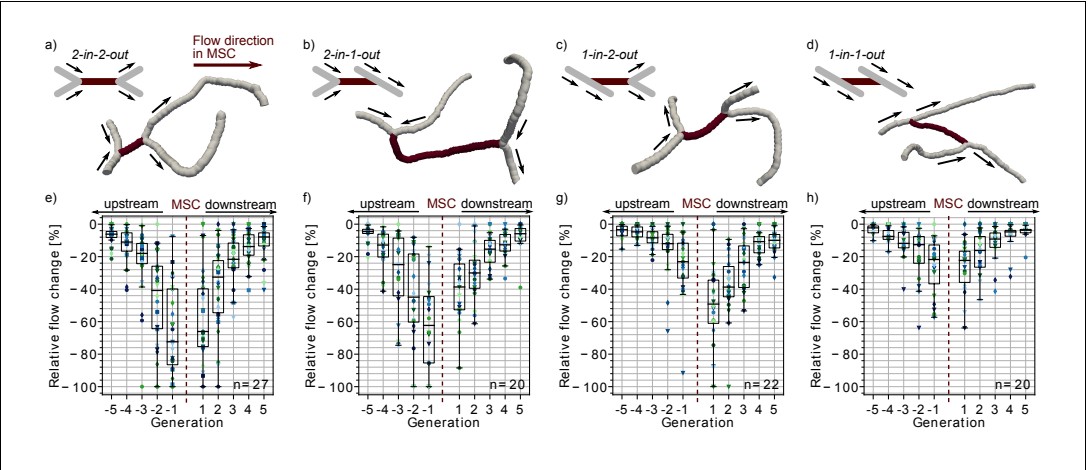

**Figure 1.** Impact of the local vascular topology on the severity of a microstroke. (**a-d**) Illustration of the four possible topological configurations at a microstroke capillary (MSC). For each topological configuration, a schematic (upper left) and a realistic example (lower right) are provided. The MSC (dark) and its adjacent vessels (gray, generation −1 and 1) are depicted. The arrows show the flow direction. (**e-h**) Average relative change in flow rate $\Delta q_{ij}$ for capillaries upstream and downstream of the MSC, which experienced a flow decrease. For each topological configuration, the flow field for ≥20 microstrokes has been computed (n: number of microstroke simulations). The average relative decrease per generation for each simulation is depicted by the color- and marker-coded symbols. The boxplots are based on the data for each generation. Statistically significant two-way interaction between MSC type and generation: upstream: F(7.8,220.42) = 7.73, p<0.001, downstream: F(7.2,203.96) = 2.24, p=0.03 (two-way mixed ANOVA). For further statistical details, see Materials and methods and *Supplementary file 1c*. The relative change for flow decreases and increases and the frequency of flow decreases per generation is shown in *Figure 1—figure supplement 1*.

The online version of this article includes the following figure supplement(s) for figure 1:

**Figure supplement 1.** Absolute flow changes in response to occlusions of different MSC types and frequency of flow decreases.

**Figure supplement 2.** Flow direction changes for the four MSC types.

the microstroke, we performed ≥20 microstroke simulations per configuration. Based on the time-averaged flow field before and after stroke, we computed the thresholded relative flow change $\Delta q_{ij}$ for each vessel (Materials and methods). Note that, the response to long-lasting stalls (>20 s) and permanent occlusions is equivalent for the observation period considered in this work.

In a first step, we analyzed the relative flow changes $\Delta q_{ij}$ in the vessels in up to five generations up- and downstream of the MSC. As more than 80% of all capillaries in the vicinity of the MSC experienced a decrease in flow (*Figure 1—figure supplement 1e–h*), the subsequent analyses focus on capillaries with a reduction in flow. *Figure 1e–g* shows that the relative changes are larger for MSCs fed by two upstream vessels and for MSCs feeding two downstream vessels. For the *worst-case scenario*, that is MSCs with a convergent bifurcation upstream and a divergent bifurcation downstream (*2-in-2-out*, *Figure 1a,e*), the median relative decrease is still <−30% at generation ±2 from the site of occlusion. In contrast for the *best-case scenario*, that is MSCs with a divergent bifurcation upstream and convergent bifurcation downstream (*1-in-1-out*, *Figure 1d,h*), the median relative change is ~15% at generation ±2 (p<0.001 at generation ±2, *Supplementary file 1c*). The differences between *2-in-2-out* and *1-in-1-out* are even more pronounced for vessels of generation ±1. Here, the median relative drop in blood flow is as large as 70% for *2-in-2-out*, while it is only 22% for *1-in-1-out* (p<0.001 at generation ±1, *Supplementary file 1c*). *2-in-1-out* and *1-in-2-out* are intermediate MSC types (*Figure 1b,c and f,g*). For example, on the upstream side, *2-in-1-out* experienced relative changes comparable to *2-in-2-out*, while on the downstream side, the trends correspond to the ones observed for *1-in-1-out* (*Supplementary file 1c*).

Analyzing the flow direction changes reveals that the occlusion of a *2-in-2-out* lead to a flow reversal in one of the two vessels at generation −1 and 1 for most MSCs (*Figure 1—figure supplement 2a*). This is plausible because from a fluid dynamical point of view, there are only two possible

outcomes for a *2-in-2-out* occlusion. The first is the observed flow reversal at generation ±1. The second would be the complete cessation of flow in generation ±1, which would lead to a larger infarct volume and thus would increase the severity of the microstroke. The cessation of flow in generation ±1 has only been observed for a small number of MSCs of type *2-in-2-out*, *2-in-1-out* or *1-in-2-out*. For some of these scenarios, a very specific topology was identified at generation ±1, where the two generation ±1 vessels are connected to the same generation ±2 vessel (*Figure 1—figure supplement 2b*). We conclude that the cessation of flow in vessels adjacent to the MSC is rare and that the less severe flow reversal is more common. At *1-in-1-out* capillaries, flow direction changes were rare, and our results across the MSC types show that flow reversal mostly occur if the MSC is fed by two inflow/feeds two outflow capillaries.

## Capillary occlusion reduces perfusion in the tissue around the MSC and causes a local redistribution of flow

To predict oxygen and nutrient supply in the tissue around the MSC, it is important to account for changes in capillaries, which are in the direct vicinity of the MSC but not directly upstream or downstream of the MSC. Therefore, we defined an analysis box around the MSC and compute its total inflow before and after stroke (*Figure 2a–e*, Materials and methods). The initial analysis box volume is set to 0.2 nl and was chosen such that each MSC fits into the initial box volume and that the box has at least five inflows. The box volume was increased progressively and the relative inflow difference has been recomputed (*Figure 2a*, Materials and methods). This analysis allows us to comment on the reduction in perfusion of a tissue volume around the MSC capillary. Moreover, it provides an estimate of the tissue volume, which is affected by a reduced perfusion in response to the microstroke.

In line with the relative flow changes in the upstream and downstream vessels (*Figure 1e–h*), we observed the largest inflow reduction for *2-in-2-out* (*Figure 2b*). For the initial box volume, that is a volume factor of 1.0, the median inflow reduction is as large as −14%. For a volume factor of 1.75, the median inflow reduction already drops to −5.1% (p=0.005, pairwise t-test with Bonferroni correction). However, it is not until a volume of 0.6 nl (volume factor: 3.0) that the median inflow reduction approaches 0%. For MSC-type *2-in-1-out*, the median inflow reduction for a volume factor of 1.0 is −13% (*Figure 2c*). The tissue around MSC-types *1-in-2-out* and *1-in-1-out* did not experience significant changes in total inflow. Here, for all volume factors, the median inflow difference is smaller than 2.5%.

Importantly, the resulting inflow reduction in the analysis box is also affected by the topological connectivity around the MSC and the redistribution of flow in response to a microstroke. These aspects become apparent if we compare the relative flow changes in vessels with different topological positions with respect to the MSC. We discern three topological positions: (1) vessels that are directly upstream and downstream of the MSC, (2) vessels that run parallel to the MSC, and (3) vessels that do not belong to the first two categories, that is distant vessels (*Figure 2f*, *Figure 2—figure supplement 1e*, Materials and methods).

*Figure 2—figure supplement 1a* shows that for up to a volume factor of 2, more than 50% of the vessels in the analysis box are directly upstream or downstream of the MSC. In these vessels, we had a significant flow reduction (*Figure 2g*). In contrast, in the vessels that run parallel to the MSC, we observed an increase in flow (*Figure 2h*). This clearly indicates that during a microstroke the flow is redistributed to pathways parallel to the MSC. However, only ~15–20% of all capillaries in the analysis box are parallel (*Figure 2—figure supplement 1b*), and consequently, we still observed an overall flow reduction in the analysis box. In the third vessel category, the distant vessels, the median relative flow difference is <2% for all volume factors (*Figure 2i*). This result confirms that the impact of a microstroke is most pronounced in vessels that are directly connected to the MSC.

Worthy of note is that for a tissue volume of 0.4 nl (i.e. volume factor = 2) ~50% of vessels in the analysis box are distant and parallel capillaries (*Figure 2—figure supplement 1a–d*). This topological configuration might be beneficial for the robustness of perfusion of the tissue volume around the MSC. Because even if the total inflow decreases in the tissue volume around the MSC, there is always a fraction of vessels within the analysis box that are not significantly affected by the microstroke (distant vessels) or that experience an increase in flow in response to the microstroke (parallel vessels). Therewith, an even larger drop in overall perfusion can be avoided and a minimum

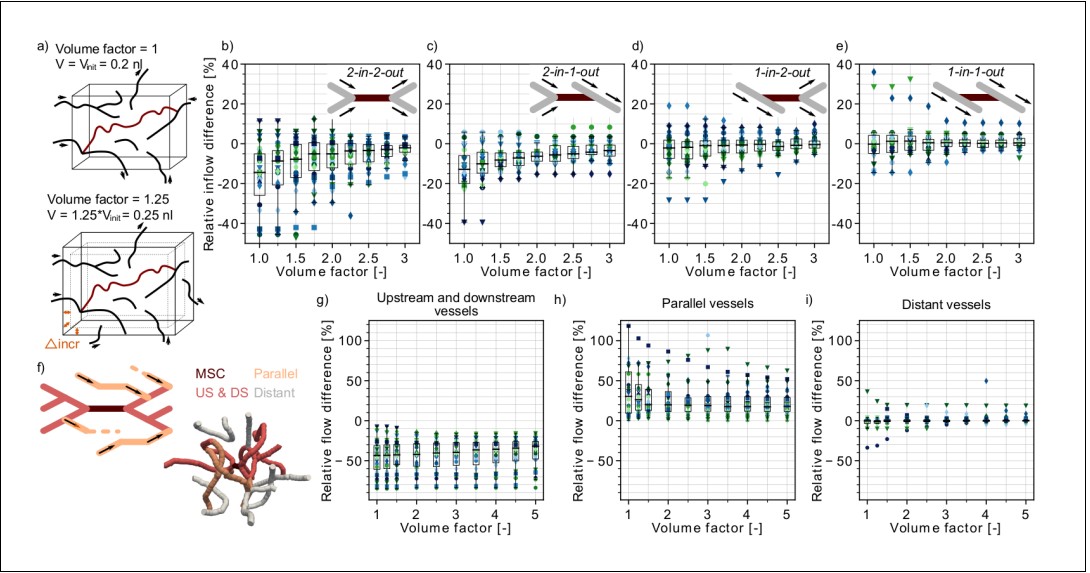

**Figure 2.** Flow reduction in analysis boxes around the microstroke capillary (MSC). (a) Schematic introducing how the volume factor is defined (Materials and methods). The MSC is highlighted in dark red. Inflow and outflow vessels of the analysis box are annotated with arrows. $V_{init}$: initial analysis box volume. $\Delta incr$: distance by which the analysis box has been increased. (b–e) Relative inflow difference for an increasing box volume around the MSC for the four MSC types. (b) *2-in-2-out*, (c) *2-in-1-out*, (d) *1-in-2-out* and (e) *1-in1-out*. The initial box volume, that is volume factor = 1, is 0.2 nl. The relative inflow difference is computed by adding up the inflows across the borders of the box for the baseline and the stroke simulation (Materials and methods). There is a statistically significant two-way interaction between MSC type and volume factor: F(5.3,150.24) = 5.23, p<0.001 (two-way mixed ANOVA). For further statistical details, see Materials and methods and ***Supplementary file 1d***. (f) Upper: Schematic to introduce the concept of vessels parallel to the MSC (Materials and methods). Lower: Realistic example of the edges in a box volume of 1 nl, that is volume factor = 5. US: upstream. DS: downstream. (g–i) Relative total flow difference for an increasing box volume around a *2-in-2-out* MSC for upstream and downstream vessels (g), parallel vessels (h), and distant vessels (i). The relative total flow difference is calculated by comparing the length-weighted flow for the baseline and the stroke simulation (Materials and methods). The ≥20 microstrokes per topological configuration are depicted by the color- and marker-coded symbols. The boxplots are based on the data for each volume factor.

The online version of this article includes the following figure supplement(s) for figure 2:

**Figure supplement 1.** Occurrences of different vessels categories within the analysis box around the MSC.
**Figure supplement 2.** Differences between the relative change in flow rate and in red blood cell (RBC) flux at a *2-in-2-out* MSC.
**Figure supplement 3.** Impact of the baseline flow rate on the severity of a microstroke in a *2-in-2-out* microstroke capillary (MSC).
**Figure supplement 4.** Impact of the cortical depth on the severity of a microstroke in a *2-in-2-out* microstroke capillary (MSC).
**Figure supplement 5.** Impact of the distance of the microstroke capillary (MSC) to the penetrating vessels on the severity of a microstroke in a *2-in-2-out*.

remaining perfusion can be ensured. This likely is beneficial to avoid a significant drop in oxygen partial pressure (pO2) in the tissue around the MSC.

The most relevant hemodynamic quantity for local pO2 is the RBC flux (***Lücker et al., 2017***). Thus, in addition to investigating changes in flow, we repeated our analyses for changes in RBC flux (***Figure 2—figure supplement 2***). As RBC flux and flow rate are related the general trends are comparable for both quantities. Interestingly, the reduction in RBC flux in the analysis box around the MSC is larger than for the flow rate (***Figure 2—figure supplement 2c,d***). This indicates that a single-capillary occlusion also affects the distribution of RBCs, which might further increase the risk of local tissue hypoxia. As our study is limited to changes in perfusion within the vasculature, further investigations resolving oxygen transport within the tissue are necessary to answer the question if a single-capillary occlusion significantly affects local tissue pO2.

## The baseline MSC flow rate increases the area of impact of a microstroke

Our results demonstrate that the local vascular topology plays a crucial role in the severity of a microstroke. To identify further structural and functional characteristics relevant to the level of flow change in response to microstroke, we repeated our analysis for eight additional cases. We looked at the impact of the baseline flow rate in the MSC (case 5), the cortical depth (cases 8–12), and the distance to the penetrating vessels (cases 6–7). An overview of the selection criteria for each vessel subset is provided in *Supplementary file 1a*. In this study, we focused on *2-in-2-out* MSCs because we expect the largest changes here.

For the case with a higher baseline flow rate, we observed that the relative change tends to be larger at generations ±3, ±4, and ±5 (paired t-test: ±3: $p<0.015$, ±4: $p<0.04$, ±5: $p<0.04$, *Figure 2— figure supplement 3a,b*). However, no significant two-way interaction between baseline flow rate category and generation and no main effect of the baseline flow rate could be detected (likely because the relative changes do not differ at generations ±1 and ±2). The impact of the baseline flow rate on the changes in the vicinity of the MSC is further supported by the analysis of the total inflow change into the analysis box around the MSC (*Figure 2—figure supplement 3c–d*), where we found a significant main effect of the baseline flow rate on the relative inflow change ($F_{(1,45)} = 13.97$, $p<0.001$, two-way mixed ANOVA). Here, the most relevant difference is that the occlusion of a capillary with a high baseline flow rate increases the volume in which a significant decrease in inflow can be observed.

No significant differences could be observed for the relative changes in upstream and down-stream capillaries for occlusions at different cortical depths or with varying distance to the penetrating vessels (*Figure 2—figure supplement 4* and *Figure 2—figure supplement 5a–b*). Regarding the analysis over cortical depth, it is important to note that the baseline flow rate of the chosen MSC has to be between 0.1 and 7.0 $\mu m^3$/ms. This selection criterion might cancel out the potential effects of the decrease in flow rate over depth (*Schmid et al., 2017*; *Guibert et al., 2010*; *El-Bouri and Payne, 2018*; *Li et al., 2019*). We observed a significant effect of the position of the MSC along the capillary path for inflow changes in the analysis box around the MSC (*Figure 2—figure supplement 5c–d*). We hypothesize that these differences are affected by the relative frequency of upstream and downstream, parallel, and distant capillaries in the analysis box (*Figure 2—figure supplement 1g–h*).

## Multi-capillary occlusions

Our results show that single-capillary occlusion affects the flow rate in multiple capillaries in the direct vicinity of the MSC. Moreover, in vivo observations suggest that capillary stalls are more likely in low flow capillaries (*Erdener et al., 2019*; *Hartmann et al., 2021*), and thus, microstrokes might accumulate in the direct vicinity of the MSC. To investigate the impact of an accumulation of capillary occlusions around the MSC, we performed simulations in which three, five, seven, and nine capillaries have been occluded in the analysis box around the MSC with a volume of 0.3 nl (volume factor = 1.5, *Figure 3a*). The simulations have been performed sequentially and in each step the two capillaries with the lowest time-averaged flow rate have been occluded for the subsequent simulation.

*Figure 3b* shows that the relative flow difference in the analysis box increases with the number of occluded capillaries. This is most apparent for the occlusion of nine capillaries, where we observed a flow decrease of ~20% in an analysis box 1.6 nl (volume factor = 8). To further analyze the perfusion changes within the analysis box, we counted the number of vessels with a flow decrease within the analysis box (*Figure 3c*). The number of vessels with a flow decrease increased with the volume factor, which underlines that the capillary occlusions also affect the perfusion in neighboring vessels. Interestingly, the number of vessels with flow decrease is smaller if more capillaries are occluded. This indicates that single-capillary occlusion causes a small flow reduction in a larger number of capillaries. In contrast, if multiple proximal capillaries are occluded a re-routing of flow occurred and a smaller number of vessels experienced a flow decrease. This is consistent with the observations in *Figure 2g–i*, where we describe a flow increase in vessels parallel to the MSC and shows how the perfusion in the capillary bed adapts to local disturbances of increasing severity.

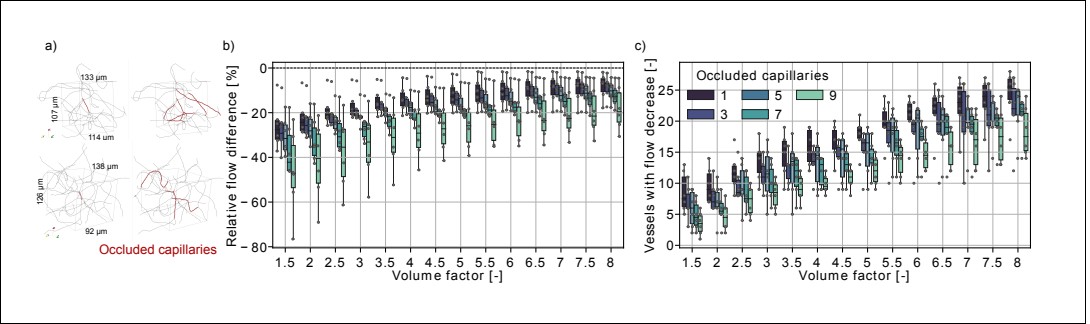

**Figure 3.** Flow reduction in analysis boxes around the microstroke capillary (MSC) for multi-capillary occlusions. (a) Capillaries in an analysis box of 1.6 nl (volume factor = 8). The occluded capillaries are highlighted in dark red (left: one occluded capillary, right: nine occluded capillaries). Two distinct examples are shown in the upper and lower row. (b) Relative total flow difference for an increasing box volume around a *2-in-2-out* MSC for an increasing number of occluded capillaries. The initial box volume, that is volume factor = 1, is 0.2 nl. The relative total flow difference is calculated by adding up the length-weighted flow for the baseline and the stroke simulation (Materials and methods). While the two-way interaction between number of occluded capillaries and volume factor is not significant ($F_{(4.92, 30.7)} = 1.06$, p=0.4), there is a statistical significant main effect of the number of occluded capillaries and the volume factor on the relative flow difference (number of occluded capillaries: $F_{(4,25)} = 3.52$, p=0.021, volume factor: $F_{(1.23, 30.7)} = 100.3$, p<0.001, two-way mixed ANOVA). (c) Number of vessels with a flow decrease in the analysis box around the MSC. Occluded capillaries are not counted. The six microstrokes per number of occlusions are depicted by the gray scatterplot. The boxplots are based on the data for each volume factor.

## The minimum distance between an arteriole-sided and a venule-sided capillary point is on average 44 μm

It is well established that pO2 in capillaries shortly downstream of DAs is higher than in capillaries just upstream of AVs (*Li et al., 2019*; *Sakadžić et al., 2014*). Moreover, it has been suggested that the tissue supplied by venule-sided capillaries might be more susceptible to hypoxia in the case of blood flow disturbances or during neural activation (*Sakadžić et al., 2014*; *Lücker et al., 2018a*; *Devor et al., 2011*). Consequently, the spatial arrangement of arteriole-sided and venule-sided capillaries with respect to each other might be an important topological feature for the robustness of oxygen and nutrient supply. A convenient way to avoid local hypoxia could be obtained by a topological structure where arteriole-sided and venule-sided capillaries are positioned in close proximity to each other.

To investigate the spatial arrangement of arteriole-sided and venule-sided capillaries with respect to each other, we introduce the *AV-factor*. The *AV-factor* for each capillary has been computed by identifying all paths leading from the capillary to all possible DA end points and to all possible AV end points. The *AV-factor* has subsequently been calculated from the median distance to all DA/ AV end points (*Figure 4a*, Materials and methods). The *AV-factor* is close to 0 if the capillary is close to a DA and is close to one if the capillary is adjacent to an AV. We define arteriole-sided capillaries as capillaries with an *AV-factor* < 0.5 and venule-sided capillaries with an *AV-factor* ≥ 0.5. The following investigations have been performed in MVN1 and MVN2. The precise analysis approaches are described in more details in the Materials and methods.

In an initial analysis we computed the shortest distance between a discretization point along a venule-sided capillary and an arteriole-sided capillary. As a reference we additionally calculated the shortest distance to any vessel around a venule-sided capillary (*Figure 4b*) and the average distance between two capillaries. The median shortest distance from a venule-sided capillary to any vessel is 18 μm and 15 μm for MVN1 and MVN2, respectively. The average distance between two capillaries is 32 μm (MVN1) and 31 μm (MVN2). The shortest distance from a venule-sided capillary to an arteriole-sided capillary is 2.73 (MVN1) and 2.78 (MVN2) times larger than the shortest distance to any vessel (*Figure 4c*). This corresponds to a distance of 46 μm (MVN1) and 41 μm (MVN2) to the closest arteriole-sided capillary, which are only factor 1.4 (MVN1) and 1.3 (MVN2) larger than the average distance between two capillaries.

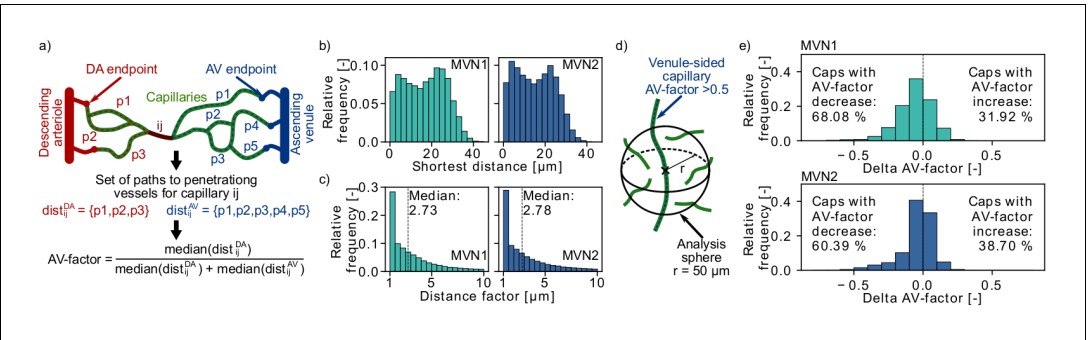

**Figure 4.** Distribution of arterial- and venule-sided capillaries within the microvascular networks (MVNs). (**a**) Schematic to introduce the concept of the AV-factor (Materials and methods). The unique flow paths from capillary ij to the descending arteriole (DA)/ascending venule (AV) main branch are annotated by p*k*. (**b**) Shortest distance to the closest vessel for venule-sided capillaries (i.e. *AV-factor* > 0.5). Each venule-sided capillary is discretized by multiple points with an average distance of 1.3 μm between the points. This results in 144,655 and 321,973 discretization points for 2753 and 7170 venule-sided capillaries in MVN1 and MVN2, respectively. The shortest distance is computed for each analysis point. (**c**) Ratio of the shortest distance from an analysis point along a venule-sided capillary to an arterial-sided capillary to the shortest distance to any vessel (=distance factor). Values with a distance factor > 10 are not displayed (15%). (**d**) Schematic to illustrate the computation of the average *AV-factor* in an analysis sphere of 50 μm around venule-sided capillaries. (**e**) Difference between the *AV-factor* of the venule-sided capillary and the mean AV-factor of the analysis spheres around the discretization points.

The online version of this article includes the following figure supplement(s) for figure 4:

**Figure supplement 1.** Average *AV-factor* in analysis cubes of varying size for microvascular network 1 (MVN1, **a**) and MVN2 (**b**).

Subsequently, we analyzed the average *AV-factor* in analysis spheres of 50 μm surrounding venule-sided capillary points (*Figure 4d*). For 68% (MVN1) and 60% (MVN2) of all venule-sided capillaries, the average *AV-factor* in the analysis sphere is smaller than the *AV-factor* of the venule-sided capillary under investigation (*Figure 4e*). This implies that these capillaries have multiple arteriole-sided points nearby, which potentially act as backup for oxygen and nutrient supply. The relative difference between the *AV-factor* of the venule-sided capillary and the mean of all points within the analysis sphere was −5.9% and −4.2% for MVN1 and MVN2, respectively. In a further analysis, we computed the average *AV-factor* for analysis cubes of different sizes (side length 30–120 μm, *Figure 4—figure supplement 1*). The range of the average *AV-factor* per analysis cube goes from almost 0 to 1. Nonetheless, the median *AV-factor* across all analysis cubes is independent of the cube size and equal to 0.52 and 0.54 for MVN1 and MVN2, respectively.

Taken together, the shortest distance of 46 μm (MVN1) and 41 μm (MVN2) to an arterial-sided capillary and the frequent decrease of the average *AV-factor* in the 50 μm analysis spheres around venule-sided capillaries suggest that arteriole-sided capillaries are well distributed throughout the network. Nonetheless, further studies are necessary to estimate whether proximal arterial-sided capillaries help to avoid hypoxic tissue areas in the vicinity of venule-sided capillaries. Moreover, it has to be kept in mind that in the rodent cortical vasculature, AVs outnumber DAs (*Schmid et al., 2019a*). This is in contrast to the primate vasculature where DAs are approximately twice as frequent as AVs (*Schmid et al., 2019a*; *Guibert et al., 2010*; *Weber et al., 2008*; *Adams et al., 2015*). As such, these results might be species dependent.

## MSC-type *1-in-1-out* supplies the largest tissue volume and is the most frequent MSC type

The significant impact of the different topological configurations on the severity of the microstroke raises questions about the frequency of occurrence and the distribution of the different MSC types in realistic MVNs. The following investigations are based on the time-averaged flow field in two realistic MVNs from the mouse somatosensory cortex acquired by *Blinder et al., 2013*, which jointly encompass a tissue volume of ~3.6 mm$^3$ and which contain 31,400 vessels (Materials and methods).

Interestingly, the *worst-case scenario*, that is MSC-type *2-in-2-out*, only occurs with a frequency of 11% (MVN1) and 6% (MVN2), while the *best-case scenario*, that is MSC-type *1-in-1-out*, represents 44% (MVN1) and 43% (MVN2) of all possible MSCs (*Figure 5d,e*). Moreover, the median-supplied tissue volume of *1-in-1-out* is 52% (MVN1) and 119% (MVN2) larger than the supplied tissue volume of *2-in-2-out* (*Figure 5f,g*, Materials and methods). Consequently, a total of 51% (MVN1) and 55% (MVN2) of the tissue is supplied by *1-in-1-out* capillaries and only 8% (MVN1) and 4% (MVN2) is supplied by *2-in-2-out* capillaries. This also becomes apparent in *Figure 5b* where the tissue volume of realistic MVN1 is color-coded based on the MSC type by which it is supplied. The differences in the median-supplied tissue volume are partly caused by the larger median vessel length of *1-in-1-out* capillaries (*Figure 5—figure supplement 1b,c*).

The small number of *2-in-2-out* capillaries and the small flow reduction for the frequent MSC-type *1-in-1-out* suggest that the cortical microvasculature is inherently robust to the occlusion of a single capillary. The significant differences in the supplied tissue volume further underline this aspect.

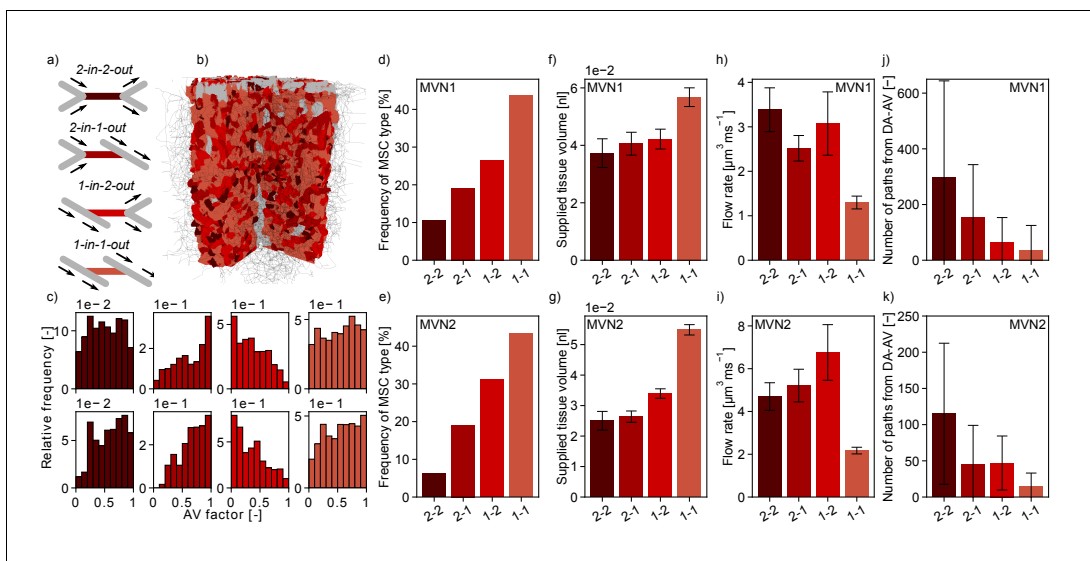

**Figure 5.** Characteristics of the four topological configurations at a microstroke capillary (MSC) for both microvascular networks (MVNs). (**a**) Schematic of the four topological configurations at a MSC. The MSC is color coded in accordance with subfigures (**b–k**). (**b**) Grid representation of the tissue in which realistic MVN1 is embedded. The tissue points are color coded based on the closest MSC type. (**c**) Relative frequency of occurrence of the different MSC types along the capillary path (*AV-factor*, Materials and methods). The number of occurrence per MSC type is normalized by the total number of capillaries per *AV-factor* bin. Upper row: MVN1 (n = 2968). Lower row: MVN2 (n = 6571). (**d, e**) Frequency of occurrence of the four MSC types (**d**: MVN1, **e**: MVN2). (**f, g**) Median-supplied tissue volume for the four MSC types (Materials and methods, **f**: MVN1, **g**: MVN2). (**h, i**) Median flow rate for the four MSC types (**h**: MVN1, **i**: MVN2). (**j, k**) Median number of paths leading through a MSC from the descending arteriole (DA) to the ascending venule (AV, **j**: MVN1, **k**: MVN2). (**f–i**) The error bars show the 95%-confidence interval. (**j, k**) The 75% confidence interval is shown. Abbreviations of the four MSC types: 2–2: *2-in-2-out*, 2–1: *2-in-1-out*, 1–2: *1-in-2-out*, 1–1: *1-in-1-out*. The statistics are based on all capillaries that fulfill the general selection criteria described in the Materials and methods. The fifth selection criterion is less strict for the current analysis, that is the capillary only has to be two segments apart from the DA/AV, and the sixth criterion is not applied. This results in 4794 and 8517 capillaries for analysis for MVN1 and MVN2, respectively. The analysis on the number of paths is based on 2968 and 6571 capillaries for MVN1 and MVN2, respectively. The Kruskal–Wallis test confirms that the differences between the MSC types are significant for the supplied tissue volume, the flow rate, and the number of paths (p<0.001 for all quantities in each MVN). p-values for pairwise comparison with the Mann–Whitney U test are listed in *Supplementary file 1e*.

The online version of this article includes the following figure supplement(s) for figure 5:

**Figure supplement 1.** Length and distance to penetrating vessels of microstroke capillaries (MSC) of different types.

**Figure supplement 2.** Median flow rate, median, and total relative supplied tissue volume for the four microstroke capillary (MSC) types over cortical depth.

*Figure 5h–i* shows that the median flow rate in a *2-in-2-out* capillary is 2.6 (MVN1) and 2.2 (MVN2) times larger than in a *1-in-1-out*. As a higher baseline flow rate increases the area of impact of a microstroke, we conclude that these differences further contribute to the severity of a microstroke in a *2-in-2-out* configuration. We hypothesize that different local topological configurations might fulfill different tasks in microvascular blood supply. While *2-in-2-out* capillaries might be more relevant for distributing blood in the cortical microvasculature, *1-in-1-out* capillaries are likely designed to robustly deliver oxygen and nutrients to the cortical tissue. This hypothesis is strengthened by the number of unique paths going from DA to AV through the different MSC types (*Figure 5j–k*, Materials and methods). While for *1-in-1-out* we only have 37 (MVN1) and 15 (MVN2) unique paths connecting DA and AV, for *2-in-2-out*, we have 297 (MVN1) and 115 (MVN2) unique paths. Generally, the described trends are consistent for MVN1 and MVN2. However, it is noteworthy that the overall average flow rate is larger in MVN2 and the number of paths per vessel is larger in MVN1. The latter is likely caused by a higher density of penetrating vessels in MVN2, which reduces the number of highly interconnected flow paths through the capillary bed.

Subsequently, we asked whether the frequency of MSC types varied over cortical depth (*Supplementary file 1b*) or along the pathway between DA and AV (*Figure 5c*, *Figure 5—figure supplement 1d–k*). The latter investigation was performed by analyzing the frequency of occurrence of the four MSC types for different *AV-factors* (*Figure 5c*). With respect to cortical depth, the frequency of the different MSC types showed the same characteristics. For the distribution of the MSC types along the pathway between DA and AV, we observed that *1-in-2-out* capillaries are more frequent on the arterial side of the capillary bed, while *2-in-1-out* are more common toward the AVs. This is plausible because at the arterial end blood is distributed to the capillary bed, while it is re-collected close to the AVs. No significant differences could be observed for the distribution of *1-in-1-out* and *2-in-2-out* capillaries along the capillary path. Notably, 93% (MVN1) and 64% (MVN2) of all paths between DA and AV contain each MSC type at least once. MSC-type *2-in-2-out* is most frequently missing along a path between DA and AV.

As previously mentioned, the median flow rate decreases significantly over cortical depth (−66% and −80% for MVN1 and MVN2, respectively). This is consistent for all MSC types (*Figure 5—figure supplement 2d–f*). No consistent trend as to how the supplied tissue volume changes over depth could be identified (*Figure 5—figure supplement 2g–i*). Important to note, is that the largest supplied tissue volume is found for *1-in-1-out* capillaries in all ALs and the relative supplied tissue volume (*Figure 5—figure supplement 2j–l*) does not vary significantly with depth. Consequently, our conclusion holds that *1-in-1-out* capillaries might be key capillaries for nutrient and oxygen discharge.

## A microstroke locally reduces the number of available flow paths

To further investigate the redistribution of flow during a microstroke, we analyzed the number of flow paths leading from DA to AV. To this end, we followed the flow downstream from the main branch of a DA until it reaches an AV main branch (Materials and methods). Importantly, due to the finite size of the MVN, various flow paths do not start at a DA or do not end at an AV. These flow paths are not considered (Materials and methods). In a first step, we computed the total number of unique flow paths between DA and AV main branch (*Figure 6b,e*). The total number of flow path during baseline in MVN1 is 139,399 and does not change significantly for single or multi-capillary occlusion. This large number highlights the interconnectivity of the capillary bed. Nonetheless, it is important to keep in mind that some flow paths only differ by one or a few vessel segments.

Subsequently, we investigated if single-capillary occlusion reduces the number of unique *DA-AV-endpoint-pairs*, that is the total number of fluid dynamically connected pairs between DA and AV end points. Our results show that in response to a microstroke new *DA-AV-endpoint-pairs* have been connected and existing *DA-AV-endpoint-pairs* have been lost (*Figure 6—figure supplement 1b*). However, with respect to the total number of *DA-AV-endpoint-pairs*, these changes are small (~0.2%). As such, our results suggest that single-capillary occlusion and the occlusion of up to nine proximal capillaries do not reduce the overall number of flow paths and the number of unique *DA-AV-endpoint-pairs* in MVN1.

To investigate changes in flow paths in the vicinity of the MSC, we count the number of paths going through a predefined capillary (*Figure 6—figure supplement 1c*) and analyzed the change in the number of unique flow paths going through: (1) capillaries upstream and downstream of the

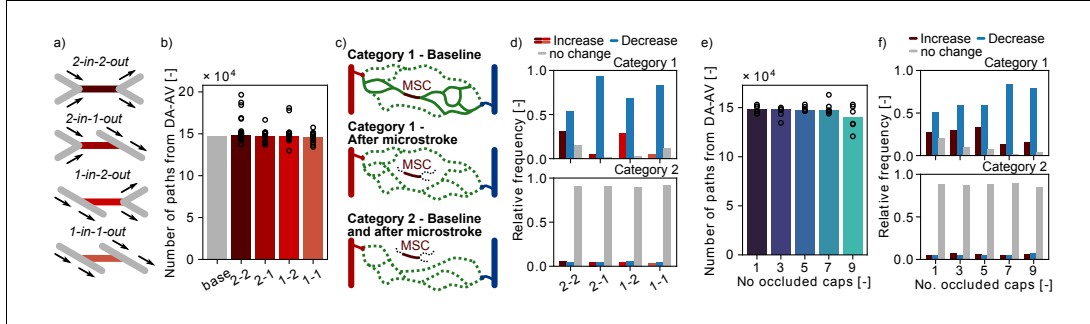

**Figure 6.** Changes in flow paths in response to single and multiple microstrokes in MVN1. (**a**) Schematic of the four topological configurations at the microstroke capillary (MSC). The MSC is color coded in accordance with subfigures (**b**) and (**d**). (**b**) Total number of flow paths connecting a descending arteriole (DA) to an ascending venule (AV) in MVN1. The bar plot depicts the results for the baseline simulation (base) and the median for the each microstroke case. The spheres show the total number of flow paths for each microstroke simulations (p=0.4, Kruskal–Wallis test). (**c**) Schematic to introduce the two categories of *DA-AV-endpoint-pairs* (Materials and methods). Each subplot shows all flow paths between one *DA-AV-endpoint-pair*. Flow paths that do not go through the MSC are labeled by the dotted line. (**d**) Relative frequency of *DA-AV-endpoint-pairs* with an increase, a decrease or no change in the number of unique flow paths for the four MSC types. The microstroke simulations per MSC type are combined before the relative frequency is computed. Upper plot: *DA-AV-endpoint-pairs* belonging to category 1. Lower plot: *DA-AV-endpoint-pairs* belonging to category 2 (see **c**). (**e**) Total number of flow paths connecting DA and an AV in MVN1 for an increasing number of occluded capillaries. The bar plot depicts the median of six simulations per number of occluded capillaries (No. occluded caps). The spheres show the total number of flow paths for each simulation (p=0.6, Kruskal–Wallis test). (**f**) Relative frequency of *DA-AV-endpoint-pairs* with an increase, a decrease or no change in the number of unique flow paths for different numbers of occluded capillaries. The microstroke simulations per number of occluded capillaries are combined before the relative frequency is computed. Upper plot: *DA-AV-endpoint-pairs* belonging to category 1. Lower plot: *DA-AV-endpoint-pairs* belonging to category 2. Abbreviations of the four MSC types: 2–2: *2-in-2-out*, 2–1: *2-in-1-out*, 1–2: *1-in-2-out*, 1–1: *1-in-1-out*.

The online version of this article includes the following figure supplement(s) for figure 6:

**Figure supplement 1.** Changes in the number of flow paths and the number of *DA-AV-endpoint-pairs* in response to a microstroke.

MSC (up to generation 3), (2) parallel to the MSC, and (3) distant to the MSC (*Figure 2f*, Materials and methods). Based on the results presented in *Figure 2g–i*, where we detected an increased flow in the parallel vessels, we expected to see an increase in the number of paths going through the parallel vessels. However, no consistent trend could be observed for the three different vessel categories (*Figure 6—figure supplement 1d*). This suggests that an increase in flow does not necessarily cause an increase in the number of flow paths through the respective capillary.

In our last analysis, we examined the number of possible flow paths between given *DA-AV-endpoint-pairs*. Therefore, the *DA-AV-endpoint-pairs* are assigned to two categories (*Figure 6c*, Materials and methods): (1) before stroke there is at least one path that leads from the DA to the AV end point through the MSC and (2) none of the paths between the given *DA-AV-endpoint-pair* go through the MSC. For *DA-AV-endpoint-pairs* of category 1, we note a decrease in the number of available flow paths between the respective *DA-AV-endpoint-pairs* for all MSC types (*Figure 6d*). The mean ratio of unique flow paths before and after microstroke at the respective *DA-AV-endpoint-pair* is between 0.67 and 0.75. If more than five capillaries are occluded in the vicinity of the MSC, the relative frequency of *DA-AV-endpoint-pairs* with a decrease in available flow paths rises (*Figure 6f*). Interestingly, the ratio of unique flow paths before and after microstroke is not affected by the number of occluded capillaries and remains at ~0.7. For the majority *of DA-AV-endpoint-pairs* of category 2, we do not observe a change in the number of unique flow path in response to single- or multi-capillary occlusion. Nonetheless, ~10% of *DA-AV-endpoint-pairs* of category 2 experience an increase or decrease in the number of connecting pathways, which underlines that the impact of capillary occlusion goes beyond directly connected vessels.

Taken together, the decrease in the number of pathways between *DA-AV-endpoint-pairs* of category 1 shows that a microstroke locally reduces the number of available paths. Based on the results shown in the preceding sections (*Figure 2h*), we suggest that the average flow rate likely increases along some of the remaining paths.

## Discussion

By performing blood flow simulations in realistic MVNs for 167 of single-capillary occlusions, we show that the severity of a microstroke depends on the local vascular topology and on the baseline flow rate in the occluded capillary. The largest impact is observed if capillaries with two inflows and two outflows (*2-in-2-out*) are occluded. Here, a drop in flow rate > 30% is still observed two generations away from the MSC. In contrast, flow rate changes remain below 25% for all capillaries at a MSC with a divergent bifurcation upstream and a convergent bifurcation downstream (*1-in-1-out*). In accordance, the occlusion of a *2-in-2-out* capillary reduces perfusion by 14% in a tissue volume of 0.2 nl. For the occlusion of a capillary with a more than three times higher baseline flow rate, a 14% drop in perfusion can still be observed for a tissue volume of 0.55 nl. Besides a local decrease in perfusion, single-capillary occlusion also causes a decrease in the number of available flow paths in the vicinity of the MSC.

Our observation that the severity of a microstroke is affected by the baseline flow rate of the occluded vessels is in agreement with previous in vivo and in silico observations for the occlusion of penetrating vessels (*Shih et al., 2013*; *Taylor et al., 2016*; *Nishimura et al., 2006*; *Blinder et al., 2013*). Additionally, *Nishimura et al., 2006* report an RBC velocity reduction in response to single-capillary occlusion of 93% and 55% in downstream vessels of generation 1–2 and 3–4, respectively. Although the in vivo velocity reductions are slightly higher, they generally compare well with our results for the occlusion of a *2-in-2-out* and a *1-in-2-out* MSC. However, *Nishimura et al., 2006* do not observe flow reversals and velocity reductions in upstream and parallel vessels. These differences are likely due to the fact that many of the occluded vessels in *Nishimura et al., 2006* are direct offshoots of the DA main branch. Due to significantly larger flow velocities in the DA main branch, velocity reductions and reversals upstream of the site of occlusion are not to be expected.

To be highlighted is that for all MSC types the effects of single-capillary occlusions are spatially constrained. To be more precise, no significant reduction in flow rate is visible five generations away from the MSC and in a tissue volume of 0.3 nl around the MSC the perfusion drops maximally by 10% for all MSC types. This is in contrast with the occlusion of DAs where the flow rate does not fully recover until the 10th downstream vessel and where the infarct volume is as large as 220 nl (*Nishimura et al., 2007*; *Shih et al., 2013*; *Blinder et al., 2013*). For scenarios in which multiple proximal capillaries are occluded, the volume of the affected area increases and the local drop in perfusion is more severe.

As the supplied tissue volume of a *1-in-1-out* MSC is 0.055 nl, which is approximately 15 times smaller than the infarct volume observed for the occlusion of a DA offshoot (*Shih et al., 2013*), we postulate that the occlusion of single-capillaries does not directly cause tissue damage. This hypothesis is in line with the results of *Shih et al., 2013*. Nonetheless, our results show that single-capillary occlusion has a strong impact on the local flow field. As such, it seems plausible that the altered flow field is a possible mechanism by which to affect A$\beta$ deposition (*Zhang et al., 2020*) or solute clearance via the perivascular space in general (*Arbel-Ornath et al., 2013*). The local disturbances in the flow field might increase the susceptibility for vessel ruptures (*van Veluw et al., 2021*) or additional occlusions, which subsequently might further impede clearance (*Hawkes et al., 2014*; *Carare et al., 2008*) as well as oxygen and nutrient supply in an increasing area around the MSC.

For the occlusion of larger caliber vessels, it has been shown that proximal microinfarcts are likely to coalesce (*Shih et al., 2013*) and that BBB leakage and intravascular platelet aggregation (*Taylor et al., 2016*), as well as deficits in neuronal activity and functional vasodynamics (*Summers et al., 2017*), are also observed beyond the microinfarct border. However, whether comparable effects can be triggered by the occlusion of a single-capillary remains unknown. Likewise, we do not know whether single-capillary occlusion induces local tissue hypoxia or if proximal vessels and increased gradients in tissue pO2 compensate for the lack of perfusion in the occluded capillary. Due to the dense capillary bed and the redistribution of flow to neighboring vessels, a single-capillary occlusion likely only causes hypoxic conditions if tissue pO2 is already low during baseline and if

the oxygen saturation in proximal capillaries is too low to compensate for the drop in perfusion. Nonetheless, because of the significant impact on the flow field, single-capillary occlusions might lead to a local drop in tissue oxygenation and oxygen saturation within RBCs, which might provoke a cascade of consecutive responses in the affected tissue around the MSC or downstream of the occluded area.

Furthermore, the impact of a microstroke on tissue oxygenation can be affected by the level of oxygen within the occluded capillary and by the local arrangement of arteriole- and venule-side capillaries with respect to each other. We hypothesize that arteriole-sided capillaries with a high oxygen content might be distributed in a convenient fashion throughout the vasculature to enhance the robustness of oxygen delivery throughout the tissue. Indeed, *Nishimura et al., 2010* provided supporting evidence for this hypothesis by showing that capillaries with varying topological distances to the DA can be in spatial proximity. However, further studies resolving oxygen partial pressure within capillaries in MVNs are necessary to confirm this hypothesis.

The remaining unknowns clearly underline the need for an in-depth in vivo quantification of the impact of single-capillary occlusion. Based on our results, we suggest that the focus of future in vivo microstroke studies should be on tissue oxygenation, Aβ deposition and long-term changes in the vicinity of the MSC. In these investigations, it is important to keep in mind that the severity of the microstroke is affected by the local vascular topology and the baseline perfusion of the MSC. Consequently, care should be taken that effects are analyzed in an MSC-type-specific manner. In addition to in vivo approaches, in silico studies resolving oxygen discharge from individual RBCs (*Lücker et al., 2015*) are a convenient tool to improve our understanding of the impact of single-capillary occlusions on tissue oxygenation.

As previously stated, the severity of a microstroke depends on the local vascular topology. Worthy of note, we observe significant differences in the frequency and the characteristics of the local vascular topologies (MSC types). *1-in-1-out* is by far the most frequent MSC type and supplies the largest tissue volume. At the same time, it is characterized by having the smallest average flow rate and by containing the smallest number of unique paths connecting DAs and AVs. In contrast, *2-in-2-out* is the rarest MSC type and contains the largest number of flow paths connecting DAs and AVs. We postulate that MSC-type *2-in-2-out* is responsible for distributing blood within the capillary bed and that MSC-type *1-in-1-out* is designed to enable efficient oxygen and nutrient discharge to the tissue. In vivo evidence supporting this hypothesis is not yet available. Here, the first step would be to confirm the characteristics of the different MSC types in vivo and to subsequently study the role of these differences on oxygen and nutrient supply.

The frequency of *2-in-2-out* MSCs is low, and in a volume of 0.2 nl around the MSC, 27% of vessels show no flow decrease after a microstroke. These two features suggest that the capillary bed offers an inherent level of robustness toward single-capillary occlusion and they agree well with the reported highly interconnected nature of the capillary bed that allows efficient re-routing of blood flow (*Schmid et al., 2019a*; *Blinder et al., 2013*; *Lauwers et al., 2008*; *Hirsch et al., 2012*; *Smith et al., 2019*; *Cassot et al., 2006*; *Lorthois and Cassot, 2010*). Nonetheless, the significant differences between the characteristics of the MSC types raise further questions regarding the origin and the severity of microstrokes.

First of all, due to the larger supplied tissue volume, might the occlusion of a *1-in-1-out* MSC be more severe for oxygen and nutrient supply, while the occlusion of a *2-in-2-out* MSC has a larger impact on the flow field? Here, in vivo studies monitoring tissue oxygenation in response to capillary occlusion or combined blood flow and oxygen transport simulations could provide insights into the most critical MSC type for oxygen and nutrient supply. Secondly, Would a microstroke be more probable in a *1-in-1-out* MSC? This idea is based on the lower average flow rate in *1-in-1-out* MSCs, which implies that the vessel might be blocked more easily (*Erdener et al., 2019*; *Hartmann et al., 2021*). However, to answer this question, we need to improve our understanding of the mechanisms that cause capillary occlusions. In healthy mice, stalls occur with a frequency < 1% (*Cruz Hernández et al., 2019*; *El Amki et al., 2020*; *Erdener et al., 2019*; *Reeson et al., 2018*). This number increases to 1.8% in AD (*Cruz Hernández et al., 2019*) and to 30% in the core of the stroke (*El Amki et al., 2020*). For both pathological conditions, the majority of stalls are caused by neutrophils adhering to the vessel wall and occurred across all capillary diameters (~4–10 µm) (*Cruz Hernández et al., 2019*; *El Amki et al., 2020*). Besides this aspect, no capillary phenotype could be identified in which stalls are more prominent (*Erdener et al., 2019*; *Reeson et al., 2018*).

Next to the phenotype of individual capillaries, the type of bifurcation might be an important factor for the likelihood of occlusion. For example, convergent bifurcations are likely more susceptible to blood particles getting stuck, which might cause an occlusion in capillaries up- and downstream of the bifurcation. However, if the origin of occlusions does not come from blocked particles but from plaque deposits or mural cell activity, then the situation is less clear and all MSC types are likely affected to similar extent.

Studying the effect of single-capillary occlusions in an isolated manner in our in silico model is advantageous on the one hand, but limited on the other hand. For example, our simulation model does not account for dynamic responses of the vasculature. It has been shown that single DA occlusion induces a heterogeneous response in the capillary bed comprising capillary dilations and constrictions (*Taylor et al., 2016*; *Nishimura et al., 2010*). Nonetheless, the in silico approach enables us to perform an in-depth study of fluid dynamical changes in response to single-capillary occlusion detached from external and internal influences. Moreover, our observations can be conveniently linked to the surrounding vascular topology. These insights can subsequently be used to distinguish changes observed in in vivo studies.

Taken together, we show that for 57% of all capillaries an occlusion significantly reduces the flow rate in the directly neighboring capillaries. Consequently, we conjecture that a single-capillary occlusion can be the starting point of a cascade of small consecutive disturbances, which might be relevant for the development of larger microinfarcts and for the progression of pathologies. In addition, we reveal novel features of the capillary bed, which are relevant for the robustness of perfusion and for advancing our understanding of topological characteristics of this highly interconnected network. Importantly, resolving the smallest scale of disturbance is not only essential to improve our understanding of microinfarct development, but might eventually offer novel possibilities for therapeutic treatment and prevention.

## Materials and methods

The presented results are based on a computational model to simulate blood flow in realistic MVNs. The model has been published previously (*Schmid et al., 2017*) and is briefly revised here. We start by giving a summary of the numerical framework to simulate blood flow in realistic MVNs with tracking of discrete RBCs (*Schmid et al., 2017*; *Schmid et al., 2015*). Subsequently, we provide more details on the analyses used in the current study.

### Blood flow modeling with discrete RBC tracking

The MVN is represented as a graph structure, that is it consists of a set of nodes $n_i$ connected by a set of edges $e_{ij}$. The subscript $ij$ indicates that edge $e_{ij}$ is connecting node $n_i$ and $n_j$. Anatomically accurate MVNs have been acquired by *Blinder et al., 2013* from the mouse somatosensory cortex by two-photon laser scanning microscopy. They are embedded in a tissue volume of ~1.6 mm$^3$ (MVN1) and ~2.2 mm$^3$ (MVN2) and contain ~12,100 and ~19,300 vessels, respectively.

The vessels are labeled as pial arteries (PAs), DAs, capillaries (Cs), AVs, and pial veins. For the penetrating vessels, that is DAs and AVs, we additionally differ between the main branch and the offshoot vessels. The vessel type is assigned by following the vessels from the cortical surface and by applying a diameter criterion which requests that two subsequent vessels have a diameter smaller than 6 μm in order to change the vessel type from DA to C (*Schmid et al., 2017*). The equivalent criterion is applied on the venule side. To differentiate between the main branch of the penetrating vessels and the offshoots, we use a criterion that is based on the branching angle and the length of the resulting main branches. This approach ensures that short offshoots are not labeled as main branch.

To compute the pressure field and the blood flow rates in the realistic MVN, we employ the continuity equation at every node and Poiseuille's law along the vessels. This approach is valid due to the small Reynolds numbers in the cortical microvasculature (Re < 1.0 for all vessels). To account for the presence of RBCs, the vessel resistance is multiplied by the relative effective viscosity $\mu_{rel}^e$. Taken together, Poiseuille's law reads

$$q_{ij} = \frac{p_i - p_j}{R_{ij}^e} = \frac{\pi D_{ij}^4}{128 L_{ij} \mu \mu_{rel}^e} (p_i - p_j)$$

where $D_{ij}$ and $L_{ij}$ are the vessel diameter and the length and $p_i$ and $p_j$ are the pressure at node $i$ and $j$, respectively. μ is the dynamic plasma viscosity and $\mu_{rel}^e$ the relative effective viscosity, which is computed as a function of the hematocrit and the vessel diameter as described in Pries et al. (in vitro formulation) (*Pries and Secomb, 2005*).

The hematocrit of individual vessels is computed from the discretely tracked RBCs. In order to correctly model the motion of RBCs, we account for the Fahraeus effect (*Pries and Secomb, 2005*; *Fåhraeus, 1929*) and the phase separation. The phase separation in vessels with a diameter larger than 10 μm is described based on the empirical relation by *Pries and Secomb, 2005*. In vessels with a diameter < 10 μm, single file flow can be assumed, and consequently, we postulate that the RBC follows the path of the largest pressure force (*Schmid et al., 2017*; *Schmid et al., 2015*; *Fung, 1973*; *Yen and Fung, 1978*). The unequal partitioning of RBCs at divergent bifurcations and their impact on the vessel resistance cause a fluctuating flow and pressure field. In the current study, we focus on the analysis of the time-averaged flow field of the statistical steady state. Our average is computed over 10 turnover times (15.4 s), where one turnover time is defined as the time until 85% of all vessels have been completely perfused at least once.

The pressure boundary conditions are assigned as described in *Schmid et al., 2017*. In brief, at the pial vessels, we make use of existing experimental data and assign a diameter-dependent pressure value. The pressure values at capillary in- and outlets are set based on the simulation results of the hierarchical boundary conditions approach. Here, the realistic MVN is implanted into a large artificial MVN. Subsequently, the flow and pressure field for a constant hematocrit is computed and the resulting pressure values are assigned as boundary conditions. The pressure boundary conditions are kept constant for each microstroke scenario.

## Microstroke simulations

The microstroke simulations are performed in MVN1. To mimic a microstroke, the diameter of a single capillary is set to 0.01 μm. For all investigated scenarios, the resulting flow rate in the MSC is $<10^{-10}\mu m^3 ms^{-1}$. The average flow rate in a capillary in MVN1 is $4.2\mu m^3 ms^{-1}$. This proves that the flow rate in the MSC approaches $0\mu m^3 ms^{-1}$ and therewith confirms the validity of our microstroke model.

In total, there are 11,386 capillaries in realistic MVN1. To ensure that we choose representative MSCs the following selection criteria have to be fulfilled:

1. The MSC should be located in a cylinder with a radius of 444 μm around the x-y-center of the MVN (number of possible MSCs: 8718).
2. The average flow rate in the MSC has to be $>0.16\mu m^3 ms^{-1}$ (95% of all capillaries, number of possible MSCs: 8237). In MVN2, this corresponds to 98% of all capillaries.
3. The average hematocrit needs to be >0.02 (95% of all capillaries, number of possible MSCs: 7824).
4. The flow rate in the MSC and its upstream and downstream neighbors should be stable, that is frequent flow direction changes should not occur. To be more precise, we allow 5%, 10%, and 30% flow direction changes in the MSC, in the first upstream and downstream vessels, and in the second and third upstream and downstream vessels, respectively (number of possible MSCs: 6307). The relative number of flow direction changes is computed by dividing the number of time steps with a flow direction change by the total number of simulated time steps.
5. The MSC is located approximately in the center of the capillary bed, that is it is at least three segments apart from the main branch of the DA and AV (number of possible MSCs: 3543).
6. The MSC has to fit into a bounding box with a volume of 0.2 nl (number of possible MSC: 3440).

It should be noted that the 'number of possible MSCs' is computed by subsequently considering an additional selection criterion.

One of our goals is to comment on factors influencing the severity of a microstroke. To analyze the impact of different factors, for example the baseline flow rate in the MSC, additional selection criteria may be prescribed. These criteria are defined in more detail in the related results sections. *Supplementary file 1a* provides an overview of all selection criteria. In total, we analyzed 12 different cases. For each case, at least 12 microstroke simulations have been performed.

## Thresholded relative change

A major challenge in comparing blood flow simulations in realistic MVNs is the large variety in flow rates, which ranges from $0.09 \mu m^3 ms^{-1}$ to values as large as $26.76 \mu m^3 ms^{-1}$ in the capillary bed (minimum and maximum of 95% of all flow rates in the capillary bed, median:$1.99 \mu m^3 ms^{-1}$). In such a flow field a large relative change in a vessel with a small baseline flow rate might be negligible, while a small relative change in a vessel with a large baseline flow rate might be significant. To improve the comparability of simulation results, we introduce an absolute threshold, $th^{abs}$. If the absolute change is smaller than $th^{abs}$ the relative change is set to 0%.

To choose an appropriate threshold value, we compare the average flow rate at two points in time for three different averaging intervals (10, 5, and 3 turnover times). The absolute flow change between the two time points is characteristic for the fluctuations of the baseline flow field. Thus, it can be used as a reference of how large the absolute flow change needs to be such that it is likely to be caused by the microstroke and not by baseline fluctuations. The difference between the two time points is 20 s.

It becomes apparent that for each of the three averaging intervals > 87% of the capillaries change their flow rate by less than $0.1 \mu m^3 ms^{-1}$ (*Supplementary file 1f*). Consequently, in the microstroke simulations a flow rate change $>0.1 \mu m^3 ms^{-1}$ is likely caused by the impact of the microstroke and not by baseline fluctuations. Accordingly, we set the absolute threshold $th^{abs}$ to $0.1 \mu m^3 ms^{-1}$.

The relative change in flow rate can either be computed directly from the flow rate in the vessel.

$$\Delta^{dir} q_{ij} = \frac{q_{ij}^{stroke} - q_{ij}^{base}}{q_{ij}^{base}}$$

or from the absolute flow rates in the vessel,

$$\Delta q_{ij} = \frac{\left| q_{ij}^{stroke} \right| - \left| q_{ij}^{base} \right|}{\left| q_{ij}^{base} \right|},$$

where $q_{ij}^{base}$ and $q_{ij}^{stroke}$ are the flow rates in vessel $ij$ for the baseline and the simulation with microstroke, respectively. Even though the second formulation neglects changes in flow direction, it is more suitable for comparing the total perfusion of individual capillaries. As the total perfusion is more relevant for oxygen and nutrient supply, we employ the second expression and analyze flow direction changes separately. Taken together the thresholded relative change is computed as

$$\Delta^{dir} q_{ij} = \begin{cases} \frac{|q_{ij}^{stroke}| - |q_{ij}^{base}|}{|q_{ij}^{base}|} & for \left| q_{ij}^{stroke} \right| - |q_{ij}^{base}| \geq th^{abs} \\ 0.0 & for \left| q_{ij}^{stroke} \right| - |q_{ij}^{base}| < th^{abs} \end{cases}.$$

The same approach is used to compute the relative change in RBC flux. Here, the threshold is set to 0.2 RBCs/s.

## Investigating differences over cortical depth

To analyze differences over cortical depth the realistic MVN is divided into five ALs each 200 μm thick (*Figure 2—figure supplement 4*). This analysis approach was first introduced by *Schmid et al., 2017*. To assign a vessel to an AL, either the source or the target vertex of the vessel has to be within the upper and lower bound of the AL (*Supplementary file 1b*). The second end point of the vessel is required to be within ±50 μm of the bounds of the AL.

## Analysis of total inflow and total flow in an analysis box around MSC

To comment on the blood supply of a tissue volume around the MSC, we compute the total inflow into an analysis box around the MSC. The volume of the smallest analysis box is chosen such that each MSC fits into the smallest analysis box (*Figure 2a*). This results in an initial box volume of 0.2 nl (200,000 μm³), which would be equivalent to a cube with a side length of 58.48 μm. Moreover, for each MSC we have at least six capillaries in the initial analysis box and at least five capillaries crossing the border of the analysis box. The chosen initial box volume is a compromise between having the smallest possible analysis box around the MSC and ensuring at the same time that sufficient

capillaries are within the analysis box to perform a quantitative investigation. The side lengths of the analysis box vary for the different MSC capillaries. To increase the box volume, the side lengths of the smallest analysis box are increased by the same distance in all three directions until we reach the desired box volume.

To compute the relative inflow change in response to a microstroke, we add up all inflows during baseline and during stroke and calculate the relative difference between the total inflow during baseline and during stroke. It is important to note that due to flow reversals in response to a micro-stroke, the number of inflow vessels can change for the baseline and the microstroke case. The equivalent analysis is repeated for increasing box volumes. The relative inflow change per analysis box is depicted in *Figure 2b–e*, *Figure 2—figure supplement 2c,d*, *Figure 2—figure supplement 3c,d*, and *Figure 2—figure supplement 5c,d*.

The change in total flow rate per analysis box is computed comparably to the inflow change in an analysis box. Here, instead of computing the total inflow during baseline and during stroke, we add up the length-weighted total flow rates in the analysis box for baseline and during stroke by summing up the flow rate of all vessels in the analysis box. We consider the vessel tortuosity to compute the vessel length within the analysis box. The total flow rate change per analysis box is used in *Figure 2g–i*.

## Definition of vessels parallel and distant to the MSC

To study the redistribution of flow in an analysis box, we introduce three vessel categories: (1) Vessels *upstream and downstream* of the MSC. (2) Vessels that branch off/into an upstream/downstream vessel of generation 1 or 2 of the MSC (*parallel vessels*). Here, we follow each *parallel vessel* of generation 1 and 2 three segments downstream/upstream to create the entire set of parallel vessels. (3) All other vessels in the analysis box (i.e. neither upstream, downstream, nor parallel vessels) are called *distant vessels*. A schematic drawing of these vessel categories is provided in *Figure 2f*, *Figure 2—figure supplement 1e*.

As the whole MVN is connected, *distant vessels* are also connected to the MSC. However, for this vessel category, the point of connection is relatively far upstream or downstream. This approach allows us to study changes in response to a microstroke with respect to the topological distance from the MSC. Note that the concept of *parallel vessels* has also been used in *Nishimura et al., 2006*. However, their definition of *parallel vessels* is different from that used in our analysis. *Nishimura et al., 2006* consider *parallel vessels* to be only those vessels that have the same source vertex as the occluded vessel.

## Multi-capillary occlusion scenarios

As we hypothesize that the occlusion of a single-capillary might trigger an accumulation of additional microstrokes around the initial MSC, we designed a simulation approach where we sequentially occluded more capillaries around the MSC. From the 27 *2-in-2-out* simulations, eight MSCs have more than 12 capillaries in the analysis box around the MSC with a volume of 0.3 nl (volume factor = 1.5). From this subset we randomly picked six MSC capillaries to study the effect of multi-capillary occlusions. Based on the time-averaged blood flow rates for the simulation with N occlusions, we chose the two capillaries with the lowest blood flow rate in the analysis box around the MSC. These two capillaries will subsequently be occluded and the time-averaged flow field will be recomputed for the new setup with N + 2 occlusions. This sequential approach is repeated until we reach nine occlusions in the analysis box of 0.3 nl around the MSC (*Figure 3a*).

## Computation of *AV-factor* and related investigations

The *AV-factor* is computed to distinguish between capillaries that are close to DAs (arteriole-sided capillaries) and those that are close to AVs (venule-sided capillaries). For each capillary *ij*, we computed all paths to all DA end points and all paths to all AV end points (*Figure 4a*). This results in a set of path lengths on the arteriole side of the capillary ($dist_{ij}^{DA}$) and a set of path lengths on the venule side ($dist_{ij}^{AV}$). To compute the *AV-factor* for capillary *ij*, we use the median of each set of path lengths, that is,

$$AV-factor_{ij} = \frac{\text{median}\left(dist_{ij}^{DA}\right)}{\text{median}\left(dist_{ij}^{DA}\right) + \text{median}\left(dist_{ij}^{AV}\right)}.$$

The resulting *AV-factor* lies between 0 and 1 and approaches 0 on the arterial side and one on the venule side of the capillary path.

To investigate the distribution of arteriole- and venule-sided capillaries within the vascular network, the following investigations have been performed. First for each venule-sided capillary, we computed the shortest distance to any vessel (*Figure 4b*) and to the closest arteriole-sided capillary (*Figure 4c*). For this analysis, each vessel is split into multiple discretization points, which are on average 1.3 μm apart and which are used to compute the shortest distance. Note that the *AV-factor* can only be assigned to capillaries along a flow path from DA to AV. Due to the finite size of our simulation domain, the *AV-factor* is only assigned to 60% and 63% of all capillaries in MVN1 and MVN2, respectively. In consequence, the calculated shortest distance to an arteriole-sided capillary might be slightly overestimated.

In the second investigation, an analysis sphere with a radius of 50 μm is moved along the discretization points of every venule-sided capillary (*Figure 4d*). All capillary points within these analysis spheres are identified and used to calculate the average *AV-factor* within the analysis spheres of the venule-sided capillary. To guarantee a representative analysis, only venule-sided capillaries are considered if at least 50% of all capillary points within the analysis spheres have been assigned an *AV-factor*.

For the third study on the distribution of *AV-factors* within the vascular network, we employ analysis cubes of varying size (side length 30–120 μm). For each cube size, the vascular network is discretized by the analysis cubes and the average *AV-factor* per analysis cube is computed (*Figure 4—figure supplement 1*). Here, an overlap of half the cube side length is used between neighboring analysis cubes. The analysis cube is only considered if it contains at least four capillaries with *AV-factor* and if at least 50% of the capillaries within the analysis cube could be assigned an *AV-factor*. The analysis cube with a side length of 30 μm contains on average 4.6 ± 0.9 (MVN1) and 4.7 ± 1.0 (MVN2) capillaries with *AV-factor*. For the analysis cube with a side length of 120 μm, we find 35.7 ± 8.9 (MVN1) and 51.0 ± 11.6 (MVN2) capillaries with *AV-factor* within the cube. This results in 7336 (MVN1) and 17,915 (MVN2) analysis cubes with a side length of 30 μm and in 1316 (MVN1) and 1700 (MVN2) analysis cubes for a side length of 120 μm.

## Computation of the topological supplied tissue volume

To comment on the infarct volume of a microstroke and for further topological studies, we compute the supplied tissue volume for each vessel. To do this, the tissue is discretized on a Cartesian grid, in which the realistic MVNs are embedded. One grid cell spans $4 \times 4 \times 4$ μm$^3$, which results in ~11.6 million grid cells for MVN1 and ~15.3 million grid cells for MVN2. Each cell center is assigned to the closest vessel. By summing over all cells assigned to one vessel, we obtain the topological supplied tissue volume per vessel. It is important to note that the topological and the effective supplied tissue volume can differ significantly (*Sakadžić et al., 2014*; *Lücker et al., 2018a*; *Lücker et al., 2015*; *Lücker et al., 2018b*). This is because of different oxygen levels along the capillary path. Consequently, for vessels with high oxygen levels, the effective supplied tissue volume is likely larger than the topological supplied tissue volume and vice versa. Nonetheless, we believe that the topological supplied tissue volume is a representative characteristic for the study of topology and perfusion-related aspects of the cortical microvasculature. Please note that for simplicity the topological supplied tissue volume is called supplied tissue volume throughout this manuscript.

## Flow paths between DAs and AVs

Flow paths between the penetrating vessels are computed by following the flow from the DA to the AV. For this investigation, a DA end point is defined as the first branch point after the main branch of the DA arteriole. The equivalent definition is used for AVs, that is the end point of an AV is the point proximal to the capillary bed and the start point of the AV is the root of the penetrating tree at the cortical surface.

To compute all paths between DA and AV, we first identify all DA and AV end points. Subsequently, for each *DA-AV-endpoint-pair*, we compute all unique connecting flow paths. Note that some DA and AV end points are not fluid dynamically connected. Additionally, multiple paths enter/ leave the MVN across its boundaries. As these paths do not connect a DA with an AV, they are not considered any further for this analysis.

The resulting flow path data allows for various investigations:

1. The computation of the total number of flow paths in the MVN (*Figure 6b,e*).
2. The computation of the number of flow paths per capillary and how this number changes during a microstroke (*Figure 6—figure supplement 1d*). We focus on the relative change in the number of paths during baseline and during stroke. This is calculated as $100\%(n_{paths}^{stroke} - n_{paths}^{baseline})/n_{paths}^{baseline}$, where $n_{paths}^{baseline}$ and $n_{paths}^{stroke}$ are the number of flow paths through an individual capillary during baseline and during stroke, respectively.
3. Instead of looking directly at the flow paths, we can also analyze the number of unique *DA-AV-endpoint-pairs*. Here, we are interested in the total number of unique *DA-AV-endpoint pairs*. As before, this quantity can be compared between the baseline and the stroke simulation. In *Figure 6—figure supplement 1b*, we compare the difference of the total number of *DA-AV-endpoint pairs* before and after stroke.
4. Lastly, we count the number of unique flow paths between a given *DA-AV-endpoint-pair* (*Figure 6c*). To comment on the redistribution of flow with respect to the MSC, we introduce two categories to classify *DA-AV-endpoint-pairs* (*Figure 6c*): Category (1) before stroke there is at least one path that leads from the DA to the AV end point through the MSC and Category (2) none of the paths between the given *DA-AV-endpoint-pair* go through the MSC. Each *DA-AV-endpoint-pair* is assigned to the according category and the ratio of the number of unique flow paths is computed $n_{DA-AV-paths}^{stroke}/n_{DA-AV-paths}^{baseline}$, where $n_{DA-AV-paths}^{baseline}$ and $n_{DA-AV-paths}^{stroke}$ are the number of unique flow paths between a given *DA-AV-endpoint-pair* during baseline and during stroke. In *Figure 6d,f*, we show the relative frequency of *DA-AV-endpoint-pairs* with an increase in the number of unique flow paths (i.e. ratio > 1), a decrease in the number of unique flow paths (i.e. ratio < 1), or no change in the number of unique flow paths (i.e. ratio = 1) in response to a single (*Figure 6d*) or multiple microstrokes (*Figure 6f*). Note that in theory there is a third category of *DA-AV-endpoint-pairs*, which are connected by a flow path through the MSC during baseline and during stroke. However, as the flow rate through the MSC during stroke is close to 0, these *DA-AV-endpoint-pairs* are excluded from the analyses.

## Statistics

Depending on the underlying data, different statistical tests have been employed. The statistical tests have been performed in R or with the Python Library *Stats*. The statistical output is summarized in the figure legends and in *Supplementary file 1c, d and e*. For the relative changes presented in *Figures 1–3* , *Figure 2—figure supplement 3* and *Figure 2—figure supplement 4*, we use a two-way mixed ANOVA with Bonferroni correction (*Supplementary file 1c-d*). As the data in *Figure 2— figure supplement 2* is paired, we compare the grouped data with the paired Wilcoxon test (grouping based on generation and volume factor, respectively). For differences in the characteristics of the four MSC types, we use the Kruskal–Wallis and the Mann–Whitney U test (*Figure 5, Supplementary file 1e*). To test for a statistical significant differences in the total number of unique flow paths (*Figure 6b,e*), we employ the Kruskal–Wallis test.

## Acknowledgements

We thank David Kleinfeld, Philbert Tsai, and Pablo Blinder for sharing the realistic microvascular networks with us. Moreover, we are grateful for the fruitful discussions of our results with Eva Erlebach, Robert Epp, and Jacqueline Condrau. Additionally, we thank Eva Erlebach for her feedback on our manuscript. We thank Karen Everett for editorial help with the manuscript.

## Additional information

### Funding

| Funder | Grant reference number | Author |
|---|---|---|
| University of Zurich | FK-19-045 | Franca Schmid |
| Horizon 2020 - Research and Innovation Framework Programme | 720270 | Franca Schmid |
| Horizon 2020 - Research and Innovation Framework Programme | 785907 | Franca Schmid |
| Swiss National Science Foundation | 310030_182703 | Bruno Weber |
| Swiss National Science Foundation | SNF CR23I2_166707 | Patrick Jenny Bruno Weber |

The funders had no role in study design, data collection and interpretation, or the decision to submit the work for publication.

### Author contributions

Franca Schmid, Conceptualization, Data curation, Software, Formal analysis, Supervision, Funding acquisition, Validation, Investigation, Visualization, Methodology, Writing - original draft, Writing - review and editing; Giulia Conti, Data curation, Formal analysis, Validation, Investigation, Visualization, Methodology, Writing - review and editing; Patrick Jenny, Bruno Weber, Resources, Supervision, Funding acquisition, Writing - review and editing

### Author ORCIDs

Franca Schmid ![ORCID] https://orcid.org/0000-0002-0689-9366
Giulia Conti ![ORCID] https://orcid.org/0000-0002-2472-1019
Patrick Jenny ![ORCID] https://orcid.org/0000-0002-9571-3104
Bruno Weber ![ORCID] https://orcid.org/0000-0002-9089-0689

### Decision letter and Author response

Decision letter https://doi.org/10.7554/eLife.60208.sa1
Author response https://doi.org/10.7554/eLife.60208.sa2

## Additional files

### Supplementary files

• Supplementary file 1. Selection criteria, MSC types over depth and statistics. (**a**) Overview of the eight selection criteria used to analyze the impact of structural and functional characteristics on the severity of a microstroke. The different microstroke capillary (MSC) types are depicted in *Figure 1a–d*. For cases 1–7, the cortical depth selection criterion requires that only the source of the MSC be within the given range. For cases 8–12, at least one of the vertices should be within the given range, while the second one may be ±50 µm outside the given range. The mean and standard deviation (std) are calculated from the results of the baseline simulation for the eight chosen MSC per case. For the mean and std of the cortical depth the values of the source and the target vertex are both considered. The definition of the main branch is provided in the methods. DA: descending arteriole. AV: ascending venule. n: simulated number of MSCs per case. (**b**) Distribution of microstroke capillary (MSC) types over cortical depth for microvascular network (MVN) 1 and 2. AL: analysis layer. Abbreviations of the four MSC types: 2–2: *2-in-2-out*, 2–1: *2-in-1-out*, 1–2: *1-in-2-out*, 1–1: *1-in-1-out*. (**c**) Statistical results for the effect of the MSC type on the changes observed at different generations (*Figure 1*). The effect of the MSC type has been analyzed separately for the generations upstream (−5 to −1) and downstream (1 to 5) of the MSC. The statistical test has been performed in R with the function anova_test() as a two-way mixed ANOVA with Bonferroni correction. Upper: There is a

significant simple main effect of the factor MSC type at all generations except generation 4 and 5. Lower table: Pairwise t-test to determine for which MSC types there is a significant difference in the changes observed per generation. Only pairs with a significant difference are listed. Case 1: *2-in-2-out*, Case 2: *2-in-1-out*, Case 3: *1-in-2-out*, Case 4: *1-in-1-out*. p-adj.: adjusted p-value, sign: significance. (**d**) Statistical results for the effect of the MSC type on the changes in inflow rate for analysis boxes of different volumes (*Figure 2b–e*). The statistical test has been performed in R with the function anova_test() as a two-way mixed ANOVA with Bonferroni correction. Upper: There is a significant simple main effect of the factor MSC type for all volume factors < 2.75. Lower: Pairwise t-test to determine for which MSC types there is a significant difference in the changes observed per volume factor. Only pairs with a significant difference are listed. Case 1: *2-in-2-out*, Case 2: *2-in-1-out*, Case 3: *1-in-2-out*, Case4: *1-in-1-out*. p-adj.: adjusted p-value, sign: significance. (**e**) Statistical results for the characteristics of different MSC types (*Figure 5f–k*). The statistical test has been performed in with the Python library scipy.stats. The Kruskal–Wallis test showed a significant difference between supplied tissue volume, flow rate, and number of paths in both microvascular networks (MVNs, all p-values<0.001). Below the p-values of the pairwise comparison with the Mann-Whitney U test are listed. Upper: p-values for MVN1. Lower: p-values for MVN2. Abbreviations for the MSC types: *2–2: 2-in-2-out*, *2–1: 2-in-1-out*, *1–2: 1-in-2-out*, *1–1: 1-in-1-out*. ns: not significant. (**f**) Absolute differences between averaged flow rates in all capillaries at two time points t1 and t2. The time difference between the two time points is 20 s. Left: The absolute differences for an averaging interval of 10 turnover times (ToT) are displayed. Middle and left: The differences for averaging intervals of 5 ToTs and 3 ToTs are shown. The absolute differences between the averaged results increase for smaller averaging intervals. For an averaging interval of 10 ToT for 94% of all vessels, the absolute difference is smaller than 0.1 $\mu m^3\ ms^{-1}$. This value decreases to 91% and 87% for an averaging interval of 5 ToT and 3 ToT, respectively.

- Transparent reporting form

## Data availability

We provide all time-averaged simulation results as well as relevant analysis scripts (see datasets table).

The following dataset was generated:

| Author(s) | Year | Dataset title | Dataset URL | Database and Identifier |
|---|---|---|---|---|
| Schmid F, Conti G, Jenny P, Weber B | 2021 | Time-averaged simulations results for bi-phasic blood flow simulations in realistic microvascular networks for various single- and multi-capillary occlusion scenarios. | https://doi.org/10.5281/zenodo.5115640 | Zenodo, 10.5281/zenodo.5115640 |

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
