## [Decision Letter]

**Acceptance summary:**

This manuscript aims to quantify the local impact of a single capillary occlusion on blood flow using in silico approaches based on realistic models of mouse microvascular networks. The authors noted four different possible arrangements of flow into and out of a capillary segment and showed that there were differing impacts on flow in up and downstream vessels for capillary occlusions with these four different arrangements.

**Decision letter after peer review:**

Thank you for submitting your article "The severity of microstrokes depends on local vascular topology and baseline perfusion" for consideration by *eLife*. Your article has been reviewed by 3 peer reviewers, and the evaluation has been overseen by a Reviewing Editor and Laura Colgin as the Senior Editor. The following individual involved in review of your submission has agreed to reveal their identity: Marilyn Cipolla (Reviewer #3).

The reviewers have discussed the reviews with one another and the Reviewing Editor has drafted this decision to help you prepare a revised submission.

Summary:

This manuscript aims to quantify the local impact of a single capillary occlusion on blood flow using in silico approaches based on realistic models of mouse microvascular networks. The authors noted four different possible arrangements of flow into and out of a capillary segment and showed that there were differing impacts on flow in up and downstream vessels for capillary occlusions with these four different arrangements. The authors then calculated the prevalence of these different arrangements in the capillary network, and later interpret these arrangements as having functionally distinct roles of blood flow redistribution vs. oxygen/nutrient exchange. The authors further calculated perfusion changes in tissue volumes due to single capillary occlusions for these different arrangements. They then examined the impact of the baseline flow rate, the depth into the cortex, and the distance from large penetrating vessels for the topological arrangement where an occlusion had the most severe impact on perfusion. The authors concluded that obstruction of capillaries that had two vessels flowing in and two flowing out had the largest impact on perfusion and that only baseline flow rate was predictive of the volume that saw reduced perfusion after occlusion of a single capillary with this two in and two out arrangement. The degree of perfusion impact suggest that single capillary occlusions do not likely lead to significant local tissue hypoxia. Finally, the authors examine how the location of a single capillary occlusion impacts the number of arteriole to venule paths through the capillary network.

Essential revisions:

General: There was a general consensus that to broaden the manuscripts appeal the authors should include a section on multiple micro-infarcts and transient stalls. The results, are somewhat narrow in scope, focusing on details of local flow rearrangements after single capillary occlusions and lacking a broader biological context or analysis of the impact of multiple occlusions (which is more relevant for disease states) that could make these findings of value to a broader scientific readership.

1. The number of cases tested for each condition is surprisingly small (eg 8 capillaries/type) and the variance in the results are calling for a much larger survey. This is similar across all the results presented. One would expect to see tens if not hundreds of cases tested for each type, across all simulations. Clearly, the data limits the amount of AD-to-AV cases but there are certainly enough capillaries to test for all the other experiments.

2a. Given the proximity of other vessels in the vicinity of the occluded capillary (some unaffected as shown by the authors, "Distant" class, Figure 2) what is the net impact of a capillary occlusion on tissue pO2? It is possible that the flow rearrangement observed by the authors (lines 231-233) counterbalance the loos of flow at the MSC keeping tissue pO2 constant (or close to) thus rendering MSC insignificant from this key physiological point of view. Therefore, this is the most crucial metric to compute as it will determine whether or not the MSC of different types do indeed lead to local ischemia; at least this should be shown for the "worst-case" scenario of 2-in-2-out. Further, it is plausible that the flow reduction in the vicinity of the MSC can allow for a larger fraction of oxygen to diffuse out of the nearby capillaries given the increased negative tissue-vessel gradient that will be generated. Combined with flow reorganization in non-affected vessels, this can lead to a net stable tissue pO2. Under such scenario, the statement in line 181-182 can be substantiated. Moreover, the relevance of MSCs is down-tuned by the reported 5% median inflow in the larger volume factor (lines 205-206).

2b. Related to the above, it is clear that this manuscript will be a cornerstone when it comes to interpreting future in vivo results. As such, the computation of tissue p02 will be extremely useful to guide such experiments.

3. With respect to the shortest distance used, the euclidean distance is of interest but the analysis should be done in terms of a "weighted" graph where resistance along the path is used. A comparison between the two is of much interest and might likely shed some interesting insight. Nevertheless, If opting for presenting one case, then the weighted one is the more relevant one.

4a. The authors have here the unique opportunity to increase the appeal of their work by including multiple-MSCs or multiple "stalls" scenarios. It is very intriguing to see in this manuscript how the presence of single MSC (and the subsequent reduction in flow in several other capillaries) increases the chances of subsequent MSC occurrence given the postulated link between initial decrease in flow (postulated as a potential mechanism for increasing the chances of MSC formation). The authors could titrate the concentration of such events based on recently published works that estimated "stall" occurrence in vivo.

4b. Is it possible for the authors to use published in vivo data to investigate what types of capillary topologies are reported with more "stalls"? Having such a comparison in this paper would be also very useful (on the same line as 2b comment).

5. There is a very interesting point made by the authors about the role of each topology (lines 544-547). This view calls for a balanced organization of the different topologies along the DA-AV flow paths. Is this indeed the case? The authors should plot the relative frequency of each type along flow paths; naively expected to be kept conserved. The opposite will weaken this view and likely point to a developmental epiphenomenon that results in a more or less random distribution of the different topologies.

6. The introduction to this manuscript is far too broad and sets up a larger problem to solve (with a focus on lack of ability to detect microstrokes in humans) for what was actually accomplished by the performed experiments. While it is good to introduce the larger picture, it is a bit misleading in terms of what the reader comes to expect after such an introduction. We recommend not only shortening it in length but also focusing it more on issues the study of flow networks are specifically able to speak to or help clarify in the field, rather than discussing uncertain implications for Alzheimer's disease pathology, for example. Perhaps emphasize the importance of investigating a single occlusion and the finding that a single occlusion is not enough to generate hypoxia likely to trigger pathology.

7. In terms of presentation of results, there was a general lack of statistical validation of trends or differences. There are several cases with large variability in simulation outcomes (e.g. the dark blue points in Figure 2f) that are not addressed. This variability and lack of statistical analysis of the results also calls into question whether the number of simulations used to generate this data was sufficient (a power analysis could prove useful here).

8. From a visual standpoint, we felt that while the results were clearly written for Figure 1, the figure itself (e to h) does not clearly show that there is a decrease in flow due to it being expressed as a relative change in flow. Figure 2 may benefit from showing a schematic representation of the volume factor, and removing the excessive horizontal gridlines (f to h). Figure 3 averages together two networks with quite different topological arrangements. It would make more sense to show the properties of these two networks independently (as in Supplementary Figure 7) in the main text. Figure 4 panel D is confusing: How can a flow path still go through the part of the vessel where a microstroke has been induced? The relevance of the decreased number of arteriole to venule flow paths after a capillary occlusion, as described in Figure 4, is also unclear. It would seem that the more biologically meaningful assay would be to explore how multiple occlusions impacted the number of flow paths and thus impacted regional perfusion. Finally, in section 3.5 a figure may be helpful to represent the finding about the geometric mixing of capillaries with different topological distances to arterioles and venules (defined as the AV-factor). It seems that this analysis could also be extended to explore how the variability of the average AV-factor of capillaries in a tissue volume varied with the size of the volume – a kind of 'smallest homogenous unit' analysis for the cortical capillary network.

9. The simulated area is small and the flow rates used also quite low. It's not clear why this was.

10. The information/data on the frequency and distribution of the different MSC-types in a realistic microvascular network is used on page 7. Where were these data obtained from?

11. The speculation that different capillary network configurations might confer distribution of blood flow and another to deliver oxygen and nutrients is interesting but not supported, yet, by data. Are there additional configurations that might be considered?

12. The authors appropriately used pressure values from Schmid that took them from published studies, however, some of those studies (Bohlen for example) used hypertensive rats. Did the authors consider normotensive or hypertensive conditions?

---

## [Author Response]

Essential revisions:General: There was a general consensus that to broaden the manuscripts appeal the authors should include a section on multiple micro-infarcts and transient stalls. The results, are somewhat narrow in scope, focusing on details of local flow rearrangements after single capillary occlusions and lacking a broader biological context or analysis of the impact of multiple occlusions (which is more relevant for disease states) that could make these findings of value to a broader scientific readership.

We agree that multiple micro-infarcts/transient stalls are more relevant scenarios for disease states. Consequently, we extended our manuscript by analysing multi-microinfarcts simulations (see Response to Q4a). The reason why we put our focus on single capillary occlusions is that we are convinced that a rigorous quantification of changes in response to single capillary occlusion is a prerequisite for understanding the role of multiple micro-infarcts, both in vivo and *in silico*. We believe that single capillary occlusions might be the starting point for accumulated flow disturbances, which likely are relevant for early disease stages and disease progression. Moreover, we want to highlight that even if microstrokes are at the heart of our manuscript our results are not limited to microstroke scenarios, but reveal novel topological and functional characteristics of the capillary bed (characteristics of different MSC-types, arrangement of arteriole and venule sided capillaries). We hope that by adding the study on multicapillary occlusions and by addressing the additional comments we were able to broaden the scope of our manuscript.

1. The number of cases tested for each condition is surprisingly small (eg 8 capillaries/type) and the variance in the results are calling for a much larger survey. This is similar across all the results presented. One would expect to see tens if not hundreds of cases tested for each type, across all simulations. Clearly, the data limits the amount of AD-to-AV cases but there are certainly enough capillaries to test for all the other experiments.

We increased the number of microstroke capillaries per case to > = 20 for the most relevant cases, i.e. the four MSC-types (case 1-4, Figure 1) and case 5 (*2-in-2-out*, high flow rate,Figure 2-figure supplement 3). For all other cases we increased the number of MSCs to >= 12. Additionally, we perform statistical tests to compare the different cases (see response to Q7), which confirm that our observations are significant.

Two additional aspects should be noted.

1. Tracking of individual red blood cells (RBCs) is computationally expensive. More precisely, one microstroke simulation takes ~15 h (single core). Therewith, the computation time for all presented results on single capillary occlusion (167 different MSCs) is 2839 h = 118 days. As such, the tested number of cases is a compromise between computational cost and significance of results. Please also not that various post-processing analyses are also computationally expensive (e.g. computation of *AVfactors* or all flow paths, shortest distance and average *AV-factor* computations) and thus limit the number of cases that can be tested.

2. We apply a set of selection criteria to chose MSCs that are representative and comparable (see Materials and methods-Microstroke Simulations and Supplementary File 1a). We believe that these selection criteria are beneficial for comparing the different cases. However, at the same time they limit the number of capillaries available for analyses.

This is most pronounced for MSC-type *2-in-2-out*, which only occurs with a frequency of 8%. Consequently, for various cases (case 5-6 and case 8-9) some selection criteria had to be relaxed to increase the number of possible MSCs (Supplementary File 1a). For example, for case 5 we had to lower the threshold for the flow rate from 7.0 µm^3^/ms to 6.6 µm^3^/ms to increase the number of capillaries fulfilling the selection criteria from 18 to 21.

2a. Given the proximity of other vessels in the vicinity of the occluded capillary (some unaffected as shown by the authors, "Distant" class, Figure 2) what is the net impact of a capillary occlusion on tissue pO2? It is possible that the flow rearrangement observed by the authors (lines 231-233) counterbalance the loos of flow at the MSC keeping tissue pO2 constant (or close to) thus rendering MSC insignificant from this key physiological point of view. Therefore, this is the most crucial metric to compute as it will determine whether or not the MSC of different types do indeed lead to local ischemia; at least this should be shown for the "worst-case" scenario of 2-in-2-out. Further, it is plausible that the flow reduction in the vicinity of the MSC can allow for a larger fraction of oxygen to diffuse out of the nearby capillaries given the increased negative tissue-vessel gradient that will be generated. Combined with flow reorganization in non-affected vessels, this can lead to a net stable tissue pO2. Under such scenario, the statement in line 181-182 can be substantiated. Moreover, the relevance of MSCs is down-tuned by the reported 5% median inflow in the larger volume factor (lines 205-206).

We completely agree that the most important metric is tissue pO2 and that our study only provides indirect evidence for changes in tissue pO2 in response to a microstroke. Both arguments provided by the reviewers (re-routing of flow and larger PO2 gradients) are mechanism, which likely are beneficial to avoid hypoxic conditions. Consequently, a simulation study accounting for tissue pO2 would be highly relevant extension of our work. However, we believe that adding a simulation study on tissue pO2 to the current manuscript is not feasible for the following reasons:

As we are interested in local effects in tissue pO2 simplified oxygen transport models,

e.g. Krogh cylinder [1], are not sufficient and an oxygen transport model resolving the vasculature and individual RBCs is necessary [2, 3]. While such models are available and have been developed and used in our lab, they are computationally expensive, e.g. simulating oxygen transport in a network of 60 capillaries for 10s takes ~40 h on 24 cores. Moreover, due to the relatively small computational domain boundary conditions (inflow haematocrits, oxygen saturation of individual RBCs, oxygen consumption of the tissue, boundary conditions of tissue domain) may affect the simulation results. Thus, in order to perform robust predictions on the effect of single capillary occlusions extensive sensitivity studies are necessary. As such, albeit its relevance, we believe that adding a study on tissue pO2 is unfortunately out of scope for the current manuscript.

Nonetheless, our study is a valuable contribution to facilitate future studies on tissue pO2. First, it provides a quantitative value for the domain size in which flow changes are to be expected. This domain size is relevant both for future in vivo and *in silico* studies. Second, the *AV-factor* (see Results 3.5) is an important aspect for local tissue oxygenation and our analysis revealed that the distribution of low and highly oxygenated capillaries might contribute to the overall robustness of oxygen supply. As such, the *AV-factor* should be considered in future analyses.

To shed more light on the role of microstrokes for tissue oxygenation, we additionally study changes in RBC flux for *2-in-2-out* capillaries. While we do not see significant differences between the relative flow and RBC flux change in the vessels up- and downstream of the MSC (Figure 2-figure supplement 2 a-b), there is a larger drop in perfusion in the analysis box around the MSC for the RBC flux (Figure 2-figure supplement2 c-d). This suggests that the single capillary occlusion also affects the distribution of

RBCs (see also lines 249-257). To highlight the relevance of tissue pO2 we altered and added various descriptions and points of discussion to the manuscript (lines 95-100, 246-247, 598-603).

2b. Related to the above, it is clear that this manuscript will be a cornerstone when it comes to interpreting future in vivo results. As such, the computation of tissue p02 will be extremely useful to guide such experiments.

We agree that the computation of tissue pO2 is a key factor to understand the consequences from single capillary occlusions. However, as stated in response to Q2a we believe that such studies should be performed thoroughly. As such, our current study is a first step in quantifying the changes in response to a microstroke, which will also provide the basis for subsequent studies on tissue oxygenation. We also want to highlight, that we believe that the presented findings are already relevant for in vivo experiments. The knowledge that the occlusion of *2-in-2-out* capillaries and of high flow capillaries causes a more severe drop in perfusion can directly be employed in in vivo experiments. Moreover, the characteristics of the different MSC-types and the arrangement of arteriole- and venule-sided capillaries are aspects, which likely are of relevance for in vivo studies focusing on perfusion characteristics of the capillary bed.

3. With respect to the shortest distance used, the euclidean distance is of interest but the analysis should be done in terms of a "weighted" graph where resistance along the path is used. A comparison between the two is of much interest and might likely shed some interesting insight. Nevertheless, If opting for presenting one case, then the weighted one is the more relevant one.

We assume that the reviewer refers to Section 3.5 where we investigated the distribution of arteriole-sided and venule-sided capillaries. Here, we used the Euclidean distance to compute the shortest distance between venule- and arteriole-sided capillary. From what we understood, the reviewer suggests to not compute the Euclidean distance between these two points but to compute the topological distance based on a weighted graph.

However, in section 3.5 our focus is on understanding the distribution of arteriole- and venule-sided capillaries from the “tissue’s perspective”. The question we ask is for example: if a venule-sided capillary is occluded, is there an arteriole-sided capillary in its proximity to delivery oxygen and nutrient to the area originally fed by the occluded capillary. For this question the Euclidean distance to the closest arteriole-sided capillary is the most relevant metric. The topological connectivity/path between the venule- and arteriole-sided capillary is in our opinion not relevant to answer this question. In fact, arteriole- and venule-sided capillary, which are close in space, might not be directly connected.

To further clarify the goal of this analysis and in response to Q8 we added a novel figure to the manuscript (Figure 4) and improved the description of this study (e.g. lines 340, 350-372, 887-914).

4a. The authors have here the unique opportunity to increase the appeal of their work by including multiple-MSCs or multiple "stalls" scenarios. It is very intriguing to see in this manuscript how the presence of single MSC (and the subsequent reduction in flow in several other capillaries) increases the chances of subsequent MSC occurrence given the postulated link between initial decrease in flow (postulated as a potential mechanism for increasing the chances of MSC formation). The authors could titrate the concentration of such events based on recently published works that estimated "stall" occurrence in vivo.

In line, with the reviewers’ suggestions and with our hypothesis that a single microstroke might cause an accumulation of microstrokes, we added a study on multicapillary occlusions, where we occlude low flow capillaries in the vicinity of the MSC (Methods and Results 3.4). The additional results show that increasing the number of occlusions does not only increase the drop in perfusion around the MSC but also increases the area of impact of the occlusion (Figure 3b). Interestingly, the number of vessels with a flow decrease in the analysis box around the MSC is smaller if more capillaries are occluded. This underlines the potential of the capillary bed to re-route blood to neighbouring vessel if local flow paths are blocked by accumulated occlusions.

We agree that our microstroke simulations are closely related to recently published work on capillary stalls. In fact, the changes for long-lasting stalls (>20s) and “permanent” microstrokes are equivalent for the observation period of 20s used in our manuscript. To highlight these similarities we extended some descriptions in the manuscript (lines 63-64, lines 132-134, lines 657-663, see also response to Q4b).

Please note that, while we do speculate that low flow capillaries are potential locations for microstrokes, our study did not focus on investigating the precise distribution of low flow capillaries within the vascular network. Moreover, for investigating the impact of single capillary occlusions we did not limit our study to low flow capillaries, but intentionally chose capillaries with an average perfusion rate. We agree that understanding the origin of capillary stalls and occlusions is an important aspect for future work. However, based on the current state of knowledge we believe that it is not possible to provide reasonable estimates for the number of capillaries prone to stalling based on our simulation results. To attain this goal, we believe the numerical model should account for the presence of neutrophils and maybe also vessel distensibility. Moreover, flow fluctuations are likely an important aspect to be considered in such an analysis.

4b. Is it possible for the authors to use published in vivo data to investigate what types of capillary topologies are reported with more "stalls"? Having such a comparison in this paper would be also very useful (on the same line as 2b comment).

We closely inspected the published in vivo work on capillary stalls to identify possible overlaps between the MSC-types as defined in our study and the locations of capillary stalls. More precisely, we looked at the work of Erdener et al. [4, 5], El Amki et al. [6], Cruz Hernandez et al. [7], Reeson et al. [8] and Santisakultarm et al. [9]. While there is some evidence that stalls are more frequent in capillaries with a lower flow rate [5], no specific vascular topology has yet been identified that increases stall prevalence and capillary stalls seem to occur across all capillary diameters, lengths and tortuosities.

Within our study we tried to investigate if the “appearance” of a capillary together with the four neighbouring capillary allows the identification the MSC-type. However, we could not identify distinct differences between the four MSC-types (see Lines 127-129), which implies that it is necessary to know the flow direction in all five vessels to accurately define the MSC-type. Thus, even if in some works the stalls are mapped to a centreline reconstruction of the local microvascular network [6, 7], without flow measurements it is impossible to identify the local MSC type.

To strengthen the link between our work and published in vivo data on capillary stalls, we extended several descriptions in the manuscript and also comment in more detail on causes and characteristics of stalling capillaries (lines 63-64, lines 132-134, lines 657663). However, further studies are necessary to answer the questions if stroke/stall prevalence varies for the different MSC-types.

5. There is a very interesting point made by the authors about the role of each topology (lines 544-547). This view calls for a balanced organization of the different topologies along the DA-AV flow paths. Is this indeed the case? The authors should plot the relative frequency of each type along flow paths; naively expected to be kept conserved. The opposite will weaken this view and likely point to a developmental epiphenomenon that results in a more or less random distribution of the different topologies.

We thank the reviewer for this valuable suggestion and added a novel figure to the manuscript (Figure 5c). Here, we plot the frequency of occurrence of the four MSC-types over the *AV-factor*. As to be expected there is a tendency that *2-in-1-out* capillaries are more frequent towards the venule side of the capillary bed, because here blood is collected to be drained via the ascending venules. The opposite trend is observed for *1-in-2-out* capillaries. *1-in-1-out* capillaries occur with the same frequency along the capillary bed, which fits well to our hypothesis that *1-in-1-out* capillaries are relevant for discharging nutrients, which needs to occur along the entire capillary path. *2-in-2-out* capillaries occur with approximately the same frequency along the capillary path (they might be slightly more frequent in the centre of the capillary bed). As such, this result is in line with out hypothesis that *2-in-2-out* capillaries might be key vessels for distributing flow.

To describe and discuss this result the manuscript has been adapted at Lines 439-447. Note that in the original manuscript we addressed this question in Supplementary Figure 5-1 d-k and Supplementary Table 3. However, we believe that the analysis via the *AV-factor* is more intuitive and provides more information on the distribution of the MSC-types along the capillary path. We kept Supplementary Figure 5-1 d-k but removed its detailed description within the manuscript to avoid duplicate results. We also removed Supplementary Table 3 from the manuscript.

6. The introduction to this manuscript is far too broad and sets up a larger problem to solve (with a focus on lack of ability to detect microstrokes in humans) for what was actually accomplished by the performed experiments. While it is good to introduce the larger picture, it is a bit misleading in terms of what the reader comes to expect after such an introduction. We recommend not only shortening it in length but also focusing it more on issues the study of flow networks are specifically able to speak to or help clarify in the field, rather than discussing uncertain implications for Alzheimer's disease pathology, for example. Perhaps emphasize the importance of investigating a single occlusion and the finding that a single occlusion is not enough to generate hypoxia likely to trigger pathology.

We thank the reviewer for pointing out that our introduction was too broad and lacked focus. We shortened the introduction by removing details on the challenges in the detection of microstrokes and by reducing details on only weakly related literature. Moreover, we try provide a more refined picture of the questions we will address and state more clearly that single capillary occlusion likely does not trigger tissue hypoxia.

Nonetheless, as single capillary occlusion is not, yet, a highly studied topic we believe it is important to provide the “bigger picture”. This includes in our opinion a brief overview on literature on the occlusion of different vessel types and more specifically on the role of capillary occlusions. We also want to point out that even though our work focuses on flow disturbances in response to single capillary occlusions alternative disturbances in response to single capillary occlusion (e.g. in tissue clearance) are possible. Here, the work by Zhang et al. [10] in an AD mouse model is a very nice example on disturbances that can be triggered by single capillary occlusions.

7. In terms of presentation of results, there was a general lack of statistical validation of trends or differences. There are several cases with large variability in simulation outcomes (e.g. the dark blue points in Figure 2f) that are not addressed. This variability and lack of statistical analysis of the results also calls into question whether the number of simulations used to generate this data was sufficient (a power analysis could prove useful here).

We performed statistical validations for all main results. Details on the statistical tests employed are found in the Materials and methods – 5.11 Statistics. The key results of the statistical tests are summarized in the Figure legends and in Supplementary File 1c-e. Note that the large variability is characteristic for the perfusion in the capillary bed and is a direct consequence of the high interconnectivity and heterogeneity within the capillary bed.

8. From a visual standpoint, we felt that while the results were clearly written for Figure 1, the figure itself (e to h) does not clearly show that there is a decrease in flow due to it being expressed as a relative change in flow. Figure 2 may benefit from showing a schematic representation of the volume factor, and removing the excessive horizontal gridlines (f to h). Figure 3 averages together two networks with quite different topological arrangements. It would make more sense to show the properties of these two networks independently (as in Supplementary Figure 7) in the main text. Figure 4 panel D is confusing: How can a flow path still go through the part of the vessel where a microstroke has been induced? The relevance of the decreased number of arteriole to venule flow paths after a capillary occlusion, as described in Figure 4, is also unclear. It would seem that the more biologically meaningful assay would be to explore how multiple occlusions impacted the number of flow paths and thus impacted regional perfusion. Finally, in section 3.5 a figure may be helpful to represent the finding about the geometric mixing of capillaries with different topological distances to arterioles and venules (defined as the AV-factor). It seems that this analysis could also be extended to explore how the variability of the average AV-factor of capillaries in a tissue volume varied with the size of the volume – a kind of 'smallest homogenous unit' analysis for the cortical capillary network.

We thank the reviewers for their suggestions on how to improve the figures of the manuscript and performed the following changes:

Figure 1: In Figure 1 we now only show capillaries that experience a flow decrease (>80% of all capillaries in the vicinity of the MSC). The exact frequency of capillaries that experience a flow decrease is available from the new Figure 1-figure supplement 1 e-h. Here, we also provide the absolute relative change if increases and decreases in flow are depicted jointly (as in the original submission). We agree with the reviewers’ that the decrease in flow was not intuitively recognizable in the original version of Figure 1. Focusing on the flow decreases highlights that this is the most common change in the vicinity of the MSC and thus underlines this result. We changed the description in the text accordingly (Lines 137-139).

Figure 2: As suggested by the reviewers we added a schematic representation of the volume factor and reduced the number of horizontal grid lines.

Figure 3 (new Figure): This is a novel figure to show the results of the multi-capillary occlusions study.

Figure 4 (new Figure): We added a novel figure to the manuscript in which we schematically introduce the concept of the *AV-factor* and show some data for the results that previously have only been described within the text. As suggested by the reviewers we also performed a ‘smallest homogenous unit’ analysis. For all analysis cube sizes the variability of the mean *AV-factor* is large and no significant impact of the cube size could be detected. The figure has been added as (Figure 4-figure supplement 1) and the results are briefly described in the text (Lines 368-372).

Figure 5 (previously Figure 3): As suggested by the reviewers we replaced Figure 5 (previously 3) with Supplementary Figure 7 of the original submission, which shows the results for both MVN independently. We agree that showing the data for the two MVNs separately provides more detailed insights. Nonetheless, we want to point out that the described trends are consistent across both networks and that differences only persist with respect to absolute values. In line with this suggestion we also adapted Figure 5-figure supplement 1 and Figure 5-figure supplement 2.

Figure 6 (previously Figure 4): We agree that our analyses on flow paths going through the MSC were confusing and did not provide very relevant insights on the flow redistribution in response to a microstroke. This is because the flow rate in the MSC is close to 0 and as such these flow paths can be neglected. Note, that we set the diameter of the MSC to 0.1 µm and consequently also for the simulation with occlusion there is a residual flow rate in the MSC (<10^-10^ 𝜇𝑚^3^𝑚𝑠^-1^), which explains why there are still flow paths going through the MSC during occlusion.

As previously, mentioned we agree that due to the flow rate close to 0 these flow paths should be neglected and removed all analyses on flow paths through the MSC from the manuscript. Instead, we now focus on the total number of unique flow paths and unique flow paths between *DA-AV-endpoint-pairs*. Moreover, we extended these analyses to the multi-capillary occlusions simulations. In line with these changes, we elaborate more clearly why we believe that the analysis of flow paths from DA to AV is relevant for understanding the distribution of flow during baseline and during stroke.

9. The simulated area is small and the flow rates used also quite low. It's not clear why this was.

Each microvascular network is embedded in a tissue volume of >1.5 µm^3^ (Lines 114 and 700). These microvascular networks (acquired by Blinder et al. [11]) are amongst the largest realistic microvascular network reconstructions that have been used successfully for simulating blood flow. Recent advances in sample preparation, imaging and network reconstruction allow acquiring whole brain microvascular networks of the mouse [1214]. However, these networks have not yet been used for blood flow simulations and it is still unknown if the accuracy of these reconstructions (mostly with respect to connectivity and diameter estimates) is sufficient for blood flow simulations. Moreover, our manuscript focuses on local changes in response to single capillary occlusion. As the area of impact of these changes is limited to a few 100 micros, it is not necessary to perform the current analysis in whole brain microvascular networks.

We are not entirely sure which flow rates the reviewer is referring to. For the blood flow simulations we do not apply any flow boundary conditions, i.e. only pressure boundary conditions are used (Lines 735-740 and response to Q12). The resulting flow field has been validated by comparison against in vivo data (see Table 3 in [15], copied in Author response table 1) and agrees well with in vivo data. The lower and upper threshold for selecting microstroke capillaries (Supplementary File 1a) has been chosen such that a large fraction of capillaries is included. In MVN1 70.3% of all capillaries have a flow rate between 0.1-4.0 µm^3^/ms and the median flow rate in the capillary bed is 1.94 µm^3^/ms. As the flow selection criterion only excludes 30% of all capillaries we are confident that the defined threshold are suitable to select characteristic capillaries.

**Author response table 1. resptable1:** Validation of the simulation results with literature data. Table from: [15].

	DA: qRBCin[nl s^-1^]	DA+A: vRBC [mm s^-1^]	C: vRBC[mm s^-1^]	C: qRBC [RBCs s^-1^]
Literature	0.1 - 10.0[50]	2.0 – 30.0[50]	mean: 0.4 – 2.0 [4, 10, 11, 13, 14, 51, 52]	mean: 38.6 – 62.0 [13, 14, 53]
MVN 1	0.88 ± 1.87	2.44 ± 4.56	0.82 ± 1.31	59.1 ± 237.6
MVN 2	5.15 ± 8.57	5.28 ± 7.61	1.38 ± 1.96	88.4 ± 574.0
MVN 3	0.96 ± 1.00	2.73 ± 4.97	0.59 ± 0.93	29.8 ± 219.0

qRBCin: RBC flow rate in the first segments of the DA, vRBC: RBC velocity in the DA+A, qRBC: RBC flux. The values of the simulation results are given as mean ± standard deviation. For the RBC flux qRBC the median ± standard deviation are given. DA: descending arteriole, A: arteriole, C: capillary, MVN: microvascular network.

10. The information/data on the frequency and distribution of the different MSC-types in a realistic microvascular network is used on page 7. Where were these data obtained from?

Our results on the frequency and distribution of different MSC-types are based on the simulation results of our time-averaged flow simulations in two realistic microvascular networks from the somatosensory cortex of the mouse. The microvascular networks have been acquired and described in detail in a previous study by Blinder et al. [11]. We provide a brief description of the computational method (originally published in [15]) and the realistic microvascular networks in the Methods. We clarified this aspect in the manuscript (Lines 397-400).

11. The speculation that different capillary network configurations might confer distribution of blood flow and another to deliver oxygen and nutrients is interesting but not supported, yet, by data. Are there additional configurations that might be considered?

On the local scale, i.e. on the scale of a single capillary, no other configurations than the four described in the manuscript are possible if trifurcations are excluded. This assumption is legit, because trifurcations only occur with a frequency ~5% (MVN1). Moreover, even though we don’t look at trifurcations, it seems likely that our observations could be extended to trifurcations, e.g. the occlusion of a *3-in-2-out* capillary would be comparable to the occlusion of a *2-in-2-out* capillary.

To the best of our knowledge, there is not yet any in vivo evidence for the role of different topological configurations for the distribution of blood/for oxygen and nutrient supply (see changes line 623-626). Nonetheless, even though this conclusion remains a hypothesis for now, the fluid dynamical characteristics of the different microvascular configurations support this (speculative) conclusion. It is also important to note, that the role of the different topological configurations is not trivial to study in vivo. Already, identifying the different MSC-types of one capillary in vivo requires five velocity measurements. To study oxygen discharge at the different MSC-types oxygen partial pressure needs to be measured in addition to the RBC velocities.

On the network scale there might be additional configurations, which might fulfil distinct functional tasks. However, we did not perform investigations on this aspect. Here, the challenge arises from the high interconnectivity of the capillary bed, which makes it difficult to identify well-defined network configurations. The arrangement of arteriole- and venule-sided capillaries (see Results – 3.5) could be considered as a configuration on the network scale that contributes to the robustness of oxygen and nutrient supply within the capillary bed.

12. The authors appropriately used pressure values from Schmid that took them from published studies, however, some of those studies (Bohlen for example) used hypertensive rats. Did the authors consider normotensive or hypertensive conditions?

Indeed, in the work of Harper and Bohlen [16] and of Werber and Heistad [17] values for normotensive and hypertensive rats are reported. To fit the diameter dependent pressure boundary conditions (as described in [15]) we used the literature data for normotensive conditions.

**References:**

1. Krogh A. The number and distribution of capillaries in muscles with calculations of the oxygen pressure head necessary for supplying the tissue. J Physiol.

1919;52(6):409-15. doi: 10.1113/jphysiol.1919.sp001839. PubMed PMID: 16993405; PubMed Central PMCID: PMCPMC1402716.

2. Lücker A, Weber B, Jenny P. A dynamic model of oxygen transport from capillaries to tissue with moving red blood cells. American Journal of Physiology – Heart and Circulatory Physiology. 2015;308:H206-H16. doi: 10.1152/ajpheart.00447.2014.

3. Lücker A, Secomb TW, Barrett MJP, Weber B, Jenny P. The Relation Between Capillary Transit Times and Hemoglobin Saturation Heterogeneity. Part 2: Capillary Networks. Front Physiol. 2018;9:1296. doi: 10.3389/fphys.2018.01296. PubMed PMID:

30298017; PubMed Central PMCID: PMCPMC6160581.

4. Erdener SE, Tang J, Kilic K, Postnov D, Giblin JT, Kura S, et al. Dynamic capillary stalls in reperfused ischemic penumbra contribute to injury: A hyperacute role for neutrophils in persistent traffic jams. J Cereb Blood Flow Metab.

2020:271678X20914179. doi: 10.1177/0271678X20914179. PubMed PMID: 32237951.

5. Erdener SE, Tang J, Sajjadi A, Kilic K, Kura S, Schaffer CB, et al. Spatio-temporal dynamics of cerebral capillary segments with stalling red blood cells. J Cereb Blood Flow Metab. 2019;39(5):886-900. doi: 10.1177/0271678X17743877. PubMed PMID:

29168661; PubMed Central PMCID: PMCPMC6501506.

6. El Amki M, Gluck C, Binder N, Middleham W, Wyss MT, Weiss T, et al. Neutrophils Obstructing Brain Capillaries Are a Major Cause of No-Reflow in Ischemic Stroke. Cell

Rep. 2020;33(2):108260. doi: 10.1016/j.celrep.2020.108260. PubMed PMID: 33053341.

7. Cruz Hernandez JC, Bracko O, Kersbergen CJ, Muse V, Haft-Javaherian M, Berg M, et al. Neutrophil adhesion in brain capillaries reduces cortical blood flow and impairs memory function in Alzheimer's disease mouse models. Nat Neurosci. 2019;22(3):41320. doi: 10.1038/s41593-018-0329-4. PubMed PMID: 30742116; PubMed Central PMCID: PMCPMC6508667.

8. Reeson P, Choi K, Brown CE. VEGF signaling regulates the fate of obstructed capillaries in mouse cortex. e*Life*. 2018;7. doi: 10.7554/*eLife*.33670. PubMed PMID:

29697373; PubMed Central PMCID: PMCPMC5919759.

9. Santisakultarm TP, Paduano CQ, Stokol T, Southard TL, Nishimura N, Skoda RC, et al. Stalled cerebral capillary blood flow in mouse models of essential thrombocythemia and polycythemia vera revealed by in vivo two-photon imaging. J Thromb Haemost.

2014;12(12):2120-30. doi: 10.1111/jth.12738. PubMed PMID: 25263265.

10. Zhang Y, Bander ED, Lee Y, Muoser C, Schaffer CB, Nishimura N. Microvessel occlusions alter amyloid-β plaque morphology in a mouse model of Alzheimer's disease. J Cereb Blood Flow Metab. 2019:271678X19889092. doi: 10.1177/0271678X19889092. PubMed PMID: 31744388.

11. Blinder P, Tsai PS, Kaufhold JP, Knutsen PM, Suhl H, Kleinfeld D. The cortical angiome: an interconnected vascular network with noncolumnar patterns of blood flow.

Nature Neurosci. 2013;16(7):889-97. doi: 10.1038/nn.3426.

12. Kirst C, Skriabine S, Vieites-Prado A, Topilko T, Bertin P, Gerschenfeld G, et al. Mapping the Fine-Scale Organization and Plasticity of the Brain Vasculature. Cell.

2020;180(4):780-95 e25. doi: 10.1016/j.cell.2020.01.028. PubMed PMID: 32059781.

13. Todorov MI, Paetzold JC, Schoppe O, Tetteh G, Shit S, Efremov V, et al. Machine learning analysis of whole mouse brain vasculature. Nat Methods. 2020;17(4):442-9. doi: 10.1038/s41592-020-0792-1. PubMed PMID: 32161395; PubMed Central PMCID:

PMCPMC7591801.

14. Ji X, Ferreira T, Friedman B, Liu R, Liechty H, Bas E, et al. Brain microvasculature has a common topology with local differences in geometry that match metabolic load.

Neuron. 2021.

15. Schmid F, Tsai PS, Kleinfeld D, Jenny P, Weber B. Depth-dependent flow and pressure characteristics in cortical microvascular networks. PLOS Computational Biology. 2017;13(2):e1005392. doi: 10.1371/journal.pcbi.1005392.

16. Harper SL, Bohlen HG. Microvascular adaptation in the cerebral cortex of adult spontaneously hypertensive rats. Hypertension. 1984;6(3):408-19. doi:

10.1161/01.HYP.6.3.408.

17. Werber AH, Heistad DD. Effects of chronic hypertension and sympathetic nerves on the cerebral microvasculature of stroke-prone spontaneously hypertensive rats.

Circulation research. 1984;55(3):286-94. doi: 10.1161/01.RES.55.3.286.

18. Shih AY, Blinder P, Tsai PS, Friedman B, Stanley G, Lyden PD, et al. The smallest stroke: occlusion of one penetrating vessel leads to infarction and a cognitive deficit.

Nature neuroscience. 2013;16(1):55-63. doi: 10.1038/nn.3278.

19. Taylor ZJ, Hui ES, Watson AN, Nie X, Deardorff RL, Jensen JH, et al. Microvascular basis for growth of small infarcts following occlusion of single penetrating arterioles in mouse cortex. J Cereb Blood Flow Metab. 2016;36(8):1357-73. doi:

10.1177/0271678X15608388. PubMed PMID: 26661182; PubMed Central PMCID: PMCPMC4976746.

20. Nishimura N, Schaffer CB, Friedman B, Tsai PS, Lyden PD, Kleinfeld D. Targeted insult to subsurface cortical blood vessels using ultrashort laser pulses: three models of stroke. Nature Methods. 2006;3(2):99-108. doi: 10.1038/nmeth844.